# Functional decomposition of metabolism allows a system-level quantification of fluxes and protein allocation towards specific metabolic functions

**Matteo Mori** [1,3] ✉, **Chuankai Cheng** [2,3], **Brian R. Taylor** [1], **Hiroyuki Okano** [1] & **Terence Hwa** [1]

Quantifying the contribution of individual molecular components to complex cellular processes is a grand challenge in systems biology. Here we establish a general theoretical framework (Functional Decomposition of Metabolism, FDM) to quantify the contribution of every metabolic reaction to metabolic functions, e.g. the synthesis of biomass building blocks. FDM allowed for a detailed quantification of the energy and biosynthesis budget for growing *Escherichia coli* cells. Surprisingly, the ATP generated during the biosynthesis of building blocks from glucose almost balances the demand from protein synthesis, the largest energy expenditure known for growing cells. This leaves the bulk of the energy generated by fermentation and respiration unaccounted for, thus challenging the common notion that energy is a key growth-limiting resource. Moreover, FDM together with proteomics enables the quantification of enzymes contributing towards each metabolic function, allowing for a first-principle formulation of a coarse-grained model of global protein allocation based on the structure of the metabolic network.

Living cells perform thousands of distinct metabolic reactions in order to grow, maintain homeostasis, and respond to environmental stimuli. Understanding the coordination of these reactions, their associated costs, and their contribution to cellular fitness are grand challenges of systems biology. Protein allocation has been established to be a key factor determining bacterial growth, owing to constraints in protein synthesis for rapidly growing bacteria[1–4]; accordingly, the abundances of proteins catalyzing these metabolic reactions have been shown to follow simple rules of allocation[5–9]. In order to integrate genome-scale data of metabolic fluxes and protein abundances, metabolic models are increasingly used as multi-omics platforms[10–15]. The inference of intracellular fluxes based on mass-balance constraints and optimization principles is a mature subject[16], with Flux Balance Analysis (FBA)

being the most celebrated framework for studying the metabolic capabilities of a wide variety of organisms[17]. Extensions of FBA have been developed to predict how global protein allocation impacts metabolic activities of the cell and vice-versa[13,18–23]. Despite the need of inferring a large number of unknown molecular parameters[24–26], these models were able to recapitulate known metabolic features, e.g. the emergence of overflow metabolism[23,27], or the global utilization of the proteome[28], in relation to cell growth.

Given the global constraint in protein synthesis, knowledge of the amount of protein needed for specific metabolic functions, e.g. the synthesis of a specific amino acid, is important for quantifying the link between metabolism and cell growth. However, the deeply interconnected nature of the metabolic networks also complicates its

[1]Department of Physics, University of California San Diego, 9500 Gilman Dr. La Jolla, San Diego, CA 92093, USA. [2]Department of Biological Sciences, University of Southern California, Los Angeles, CA 90089, USA. [3]These authors contributed equally: Matteo Mori, Chuankai Cheng. ✉e-mail: mamori@ucsd.edu

decomposition into individual components[29]. For example, central carbon pathways such as glycolysis and the TCA cycle are not only tasked with the production of metabolic precursors for biomass building blocks, but also with balancing the currency metabolites, e.g. ATP and NAD(P)H[30], consumed by each biosynthetic pathway. Thus, it can be difficult to associate reaction fluxes, and the corresponding enzyme concentrations, to individual metabolic functions. Indeed, the intuitive notion of individual metabolic pathways is largely concocted, as the production and consumption of currency metabolites have to be balanced across conditions, thus effectively coupling the fluxes through all pathways. Thus, the calculation of costs and yields for the production of individual metabolites are often performed by making use of coarse-grained metabolic networks in which the balance of currency metabolites or pathway byproducts is simplified or neglected[31,32].

Here, we sought to develop a systematic computational method to define the metabolic costs and the enzyme amounts associated with each metabolic function, thus cutting through the complexity of the network. We introduce a Functional Decomposition of Metabolism (FDM) based on the decomposition of metabolic fluxes into a set of flux components, each associated with a metabolic function. Being based on properties of optimal flux pattern such those obtained with FBA, FDM is generally applicable to any metabolic network, and does not require additional parameters. We applied FDM to the study of *Escherichia coli* cells grown in carbon minimal media, under translational-limiting antibiotics, and in anaerobic growth. The resulting functional characterization of the metabolic reactions allowed us to analyze in detail how cells allocate nutrients towards biosynthesis and energy generation, as well as the metabolic costs and yields of the production of different biomolecules. Together with experimental protein abundances, FDM allowed us to quantify the total amount of enzymes allocated to each function. Finally, FDM enabled a genome-wide classification of the proteome according to metabolic function, and the formulation of a coarse-grained model of protein allocation which quantitatively captures the global changes of the proteome across conditions.

## Results

### Functional decomposition of metabolic fluxes

In silico genome-scale models of metabolism (GEMs) enable the quantitative modeling of cellular metabolism by including information on thousands of cellular metabolites and reactions, as well as on the cellular biomass composition and the enzyme-reaction assignment[33]. The prototypical use of GEMs is the inference of intracellular fluxes by using a combination of empirical constraints and optimization, an approach that is generally termed Flux Balance Analysis (FBA)[17]. In brief, growing cells accumulate biomass building blocks (e.g. amino acids and nucleotides) at rates set by the cellular biomass composition and the growth rate, while also regenerating the ATP required for homeostasis and growth (the so-called maintenance energy flux) (illustrated in Fig. 1a, top diagram). Both biomass-associated and energetic demand fluxes are fixed empirically by the biomass composition and the fluxes of metabolite uptake or excretion. Then, the intracellular fluxes can be estimated by taking the flux pattern(s) that maximize a given objective function. Such optimal flux patterns are generally in good agreement with experimental data on intracellular

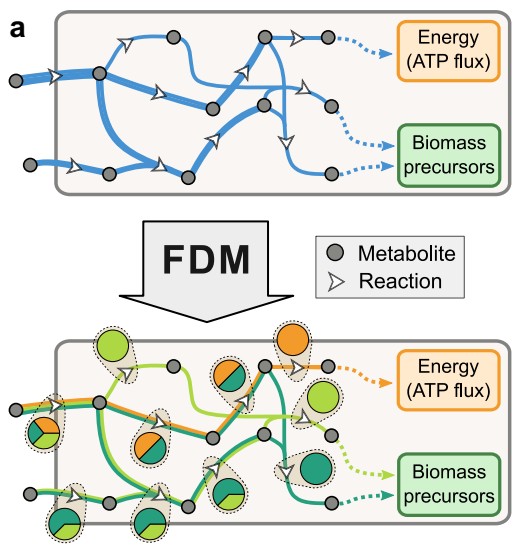

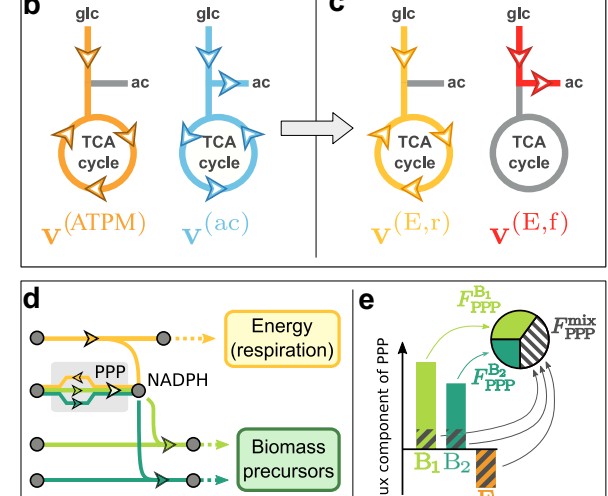

**Fig. 1 | General approach to the functional decomposition of metabolic fluxes.**
**a** Functional Decomposition of Metabolism (FDM) is a method to decompose metabolic fluxes **v** (top diagram, in blue) estimated via Flux Balance Analysis (FBA) or similar methods into several components **v**$^{(y)}$ (bottom diagram, colored lines). In this simple example, fluxes are split into three components: energy production (orange) or the biosynthesis of two biomass precursors (shades of green). These flux components allow to quantify for each reaction (white arrows) the fraction of flux that is associated with each metabolic function, indicated here by the pie charts. **b** The application of this method to *E. coli* cells grown aerobically in minimal glucose media requires overcoming two main problems. First, the presence of a constraint on acetate excretion, necessary to obtain a good agreement with experimental data, requires the presence in the flux decomposition of an acetate-associated component **v**$^{(ac)}$ (cyan) which has no clear interpretation in terms of biological functions. Second, the flux components associated with TCA reactions are found to have both positive terms, e.g. in the energy-associated component **v**$^{(ATPM)}$, and negative terms, e.g. in the acetate-associated component. This sign mismatch does not allow us to define the functional shares of these reactions as ratios of flux components, $v^{(y)}/v$ (see text). **c** By linearly combining the flux components associated with energy and acetate production, it is possible to define two new flux components associated with energy production via respiration (**v**$^{(E,r)}$, yellow) and fermentation (**v**$^{(E,f)}$, in red). These flux components have a clear biological interpretation, and do not lead to inconsistent flux directions for TCA reactions. **d** In some cases, sign mismatch among flux components is an inevitable consequence of the network structure. In this simplified example, NADPH is produced along with energy (e.g. by the ICD reaction in the TCA cycle) and consumed in biosynthetic reactions. In order to preserve a steady concentration of NADPH, the flux through the Pentose Phosphate Pathway (PPP) has to adjust upon changes in energetic or biosynthetic cellular demands, thus leading to negative flux components. **e** A functional decomposition can be defined even in presence of negative flux components. Negative components partially cancel out the positive components; the magnitude of the cancellation defines a mixed fraction $F^{mix}$ not associated with any specific biological function. The rest of the flux components are then associated with the remainder, $1 - F^{mix}$ (see Supplementary Note 2.3 for details).

fluxes for carbon-limited growth[34]. Our application of FBA to *E. coli* cells, with a detailed description of all constraints, is given in Supplementary Note 1.

Given a flux pattern obtained via FBA, we sought a general method to quantify how much a given metabolic reaction contributes to another metabolic process $\gamma$, e.g. how much of the carbon intake flux is used for the production of a given amino acid, or how much of the flux through a given glycolytic reaction is associated with the regeneration of ATP. Answering these questions is tantamount to expressing the metabolic fluxes in the network in terms of the consumption (demand) fluxes of biomass building blocks or the production of energy in each growth condition.

It is possible to provide an explicit expression that relates the FBA-derived flux vector **v**, which has entries $v_i$ for each reaction $i$, to the demand fluxes $J_\gamma$. The set of the demand fluxes appearing in such expression is not arbitrary and it depends on the specific FBA formulation used to model the metabolic fluxes. As discussed in depth in Supplementary Note 2, each non-dimensionless constraint applied to the network must be associated with a demand flux; In the case shown in Fig. 1, demand fluxes are associated with the synthesis fluxes of each biosynthetic building block and the maintenance ATP flux. (We will discuss the effect of additional constraints later below.) Because of the linear properties of the optimization problem, and as long as the flux solution is unique, the nonzero fluxes can be expressed[23] as a linear combination of the demand fluxes $J_\gamma$:

$$\mathbf{v} = \sum_\gamma \boldsymbol{\xi}^{(\gamma)} J_\gamma. \tag{1}$$

This expression represents a parameterization of the optimal fluxes in terms of the demand fluxes $J_\gamma$, and allows to analyze how the flux **v** changes in response to perturbations of the demand fluxes. The terms $\boldsymbol{\xi}^{(\gamma)}$ determine how variations in the demand fluxes $J_\gamma$ affect each reaction. To determine these coefficients we note that they match the derivatives of the fluxes with respect to the demand fluxes. Thus, they can be obtained numerically by computing the optimal fluxes upon a small perturbation of each demand flux $J_\gamma$. Taken to face value, Eq. (1) suggests that the flux pattern **v** can be partitioned into the sum of several flux components

$$\mathbf{v}^{(\gamma)} \equiv \boldsymbol{\xi}^{(\gamma)} J_\gamma, \tag{2}$$

where each component $\mathbf{v}^{(\gamma)}$ satisfies the mass-balance constraints of the network and is associated with a single demand flux $J_\gamma$. For example, if $\gamma$ represents the production of the amino acid glutamine, then both $\boldsymbol{\xi}^{(\gamma)}$ and $\mathbf{v}^{(\gamma)}$ represent a complete pathway transforming the carbon and nitrogen sources to glutamine; the two only differ by an overall normalization factor. We will use this example to illustrate our results later below.

In the context of metabolic networks, the production of biomass and energy represent natural definitions of biological functions which the cell has to perform in order to survive and grow. This offers a biological interpretation of the linear relation, Eq. (2), between cellular and demand fluxes: it represents a functional decomposition of the metabolic fluxes in which each reaction $i$ contributes to the function $\gamma$ (with associated demand flux $J_\gamma$) by a fraction $F_i^{(\gamma)} \equiv v_i^{(\gamma)}/v_i$ of the total flux $v_i$. This is illustrated in Fig. 1a, where the flux of each active reaction (blue in the top diagram) is split into several components (in different colors in the bottom diagram), and therefore to each reaction (arrows) is assigned a breakdown into different biological functions (piecharts).

We term the definition of function-specific shares for individual metabolic reactions (and, later, metabolic enzymes) Functional Decomposition of Metabolism (FDM). In the next sections we will discuss the application of FDM to the concrete case of exponentially growing *E. coli* cells in carbon minimal media. However, we will first

highlight two challenges that emerge when applying the method on realistic networks, and how both issues are solved by adjusting the set of biological functions used to define the functional decomposition.

## Additional constraints and mixed flux components

While the simple example in Fig. 1a illustrates the general idea of the method, it does not capture two important obstacles that prevent the straightforward application of FDM to realistic networks. Firstly, additional constraints are often needed to correctly model the cellular fluxes. Such additional constraints have to be accounted for in the flux decomposition, Eq. (1), but the biological interpretation of the corresponding flux components can be opaque. For example, acetate is excreted by fast-growing *E. coli* cells even in presence of oxygen, but FBA fails to model overflow metabolism without additional constraints. The simplest approach to improve FBA's ability to model the intracellular fluxes is to set the acetate production to a non-zero value $J_{ac}$ determined experimentally across growth conditions. This constraint leads to the appearance in Eq. (1) of a flux component $\mathbf{v}^{ac} = \boldsymbol{\xi}^{ac} J_{ac}$ (Fig. 1b, cyan). However, inspection of this component reveals that it does not contribute to either the production of energy or biomass components. As a result, its interpretation in terms of biological functions is not immediately apparent. Moreover, a deeper examination of the flux component revealed a second issue: reactions belonging to the TCA cycle presented negative entries $v_i^{(ac)}<0$ (indicated in figure by the counterclockwise arrows), corresponding to carbon fixation. This is opposite to the positive direction of the overall fluxes, corresponding to that of the oxidative TCA cycle (Fig. 1b, orange, clockwise arrows). Such sign mismatch is not only difficult to interpret from a biochemical standpoint, but also hinders a consistent functional decomposition for the TCA reactions: a ratio $v_i^{(\gamma)}/v_i<0$ can hardly be interpreted as the share $F_i^{(\gamma)}$ of flux associated with the function $\gamma$.

Fortunately, the linear structure of Eq. (1) allows us to couple different biological functions, expressing the flux vector as a different linear combination of flux components (see Supplementary Note 2). In this example, the negative components in the TCA flux can be eliminated by linearly combining the energy- and acetate-associated components into a new component $v_i^{(E,f)}$ (Fig. 1c, red) which describes aerobic fermentation, leading to new vectors with no negative entries and a clear biological interpretation. The example of acetate excretion shows how negative flux components can be an indication that two metabolic activities (e.g. energy generation and acetate production) are tightly coupled, and it is more sensible to consider them as a unique process (e.g. fermentation). However, not all sign-mismatched components can, or need, to be removed by coupling different biological functions. In fact, this phenomenon is generically expected in association with intermediate metabolites which are differentially produced or consumed in association to different metabolic functions. This is illustrated in Fig. 1d where we show a simplified picture of the cellular balance of NADPH. The electron donor NADPH is consumed in biosynthetic processes, and produced during aerobic respiration (by isocitrate dehydrogenase). The flux through the Pentose Phosphate Pathway (PPP) is regulated to keep the concentration of NADPH constant upon variations in the energetic and biomass-associated fluxes[35]. This particular role for PPP is reflected in the sign-mismatch among the flux components associated with energy and biomass biosynthesis. To account for such role, we introduced an additional mixed function, proportional to the amount of cancellation between positive and negative flux components:

$$F_i^{mix} = \frac{\sum_\gamma |v_i^{(\gamma)}| - |v_i|}{\sum_\gamma |v_i^{(\gamma)}|} \tag{3}$$

This quantity is equal to zero in absence of negative flux components, and approaches one in presence of large flux components of opposite

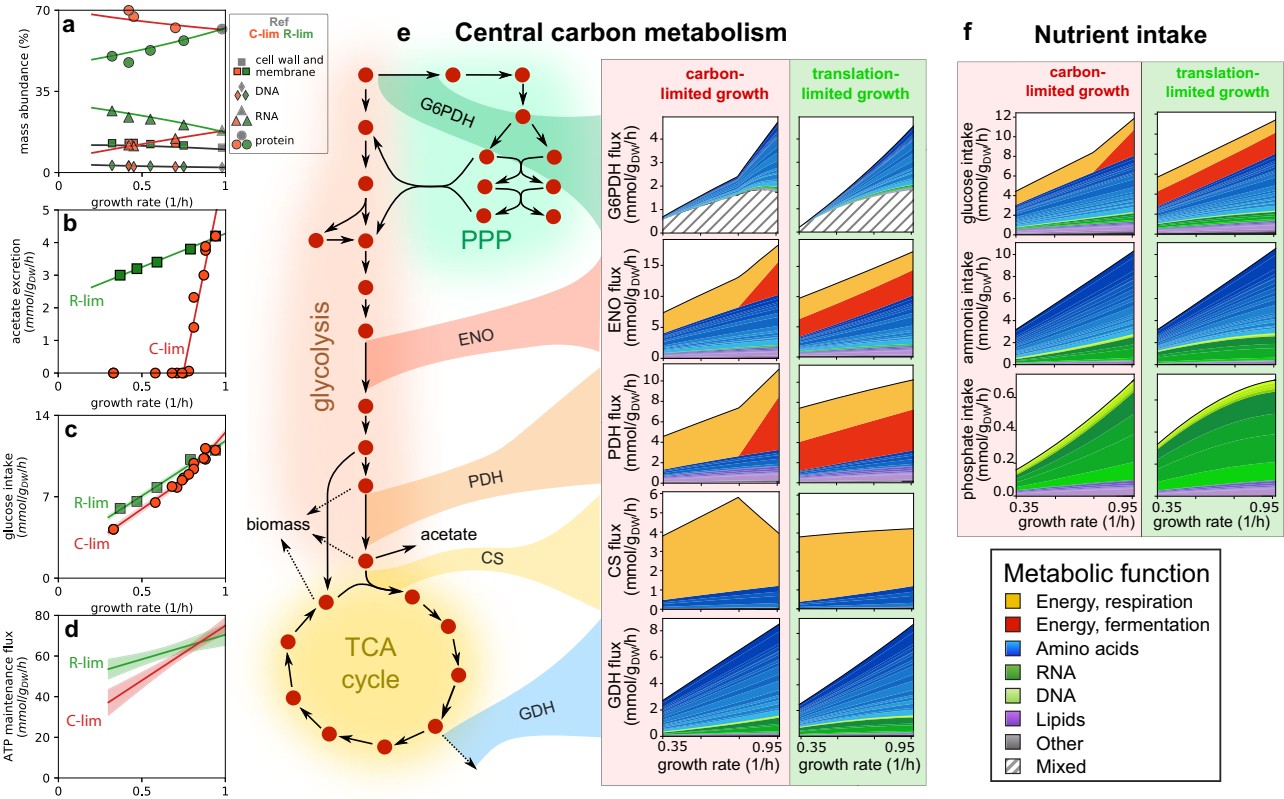

**Fig. 2 | Application of FDM to carbon- and translation-limited *E. coli* cells.**
**a** Summary of the growth-dependent biomass composition of carbon-limited (red) and translational inhibited (green) *E. coli* cells. See Supplementary Fig. 1 for a detailed breakdown of the biomass composition. **b** Acetate excretion rate against growth rate in C-limitation (red) and R-limitation (green). The solid lines are best fit to the data, and have been used to prescribe the acetate demand flux $J_{ac}$ as a function of growth rate in the two growth limitations. **c** Glucose uptake rate. Filled circles and squares indicate measurements (see Methods). Solid lines and shaded areas indicate the flux prediction (mean and 68% confidence bands) corresponding

to the best fitting ATP maintenance flux parameters (see next panel). **d** The ATP maintenance flux is modeled as a linear function of the growth rate $\mu$, $J_{ATPM} = \sigma_0 + \sigma \cdot \mu$. The solid lines and shaded areas indicate the best fit relation (mean and 68% confidence bands; parameter values in Supplementary Table S2). **e** Flux decomposition of the central carbon metabolic pathways across growth rates for C-limited (left) and R-limited (right) conditions. The reactions shown are G6P dehydrogenase (G6PDH), enolase (ENO), pyruvate dehydrogenase (PDH), citrate synthase (CS) and glutamate dehydrogenase (GDH). **f** Flux decomposition of glucose, ammonia and phosphate intake fluxes.

signs. As shown in Fig. 1e, we associated the remainder of the functional share $F_i^{(y)}$ to the predominant metabolic functions (see also Supplementary Note 2). This allowed us to consistently define a genome-wide functional decomposition $F_i^{(y)}$ even in presence of sign-mismatched flux components.

**Application to *E. coli* growing on glucose minimal medium**
To model the intracellular fluxes, we made use of the most recent *E. coli* genome-scale model of metabolism, iML1515[13], including 2719 reactions and associations to 1515 protein-coding genes. We assembled physiological data for exponentially growing *E. coli* K-12 NCM3722 cells across conditions, including a reference condition corresponding to glucose minimal media, and slower growth conditions attained by either titrating the uptake of glucose (C-limitation), or by inhibiting protein synthesis using sublethal doses of chloramphenicol (R-limitation). (See Methods and Supplementary Table S1 for the strains used in this study). For each of these conditions, we used the macromolecular composition of the cell (Fig. 2a and Supplementary Fig. 1), and determined the demand fluxes for individual amino acids, nucleotides, lipids and other small molecules (Supplementary Fig. 1). In particular, the abundance of cysteine and glycine residues were found to be quite different compared to the values in the default iML1515 model across all conditions (Supplementary Fig. 1k). Three versions of the iML1515 model, with modified biomass composition tuned to specific growth conditions, are available in Supplementary Data 1.

For each growth condition, the metabolic model was constrained using the condition-specific biomass composition to set the demand flux of each biomass building block (Methods, Supplementary Data 2); the measured acetate excretion fluxes (Fig. 2b); and the ATP maintenance flux. The latter was estimated separately for the two growth limitations by matching the minimal glucose intake allowed by the metabolic model to the measured glucose intake flux (Methods), shown in Fig. 2c. This allowed us to define maintenance fluxes specific for each of the two growth limitations explored (Fig. 2d). Finally, the metabolic fluxes were then computed with parsimonious FBA, by first minimizing the glucose uptake, and then minimizing the $L_2$-norm of the fluxes, which guarantees the uniqueness of the solution (see Supplementary Note 1). This approach allows to model the intracellular fluxes in *E. coli* with remarkable accuracy[36].

We then turned to the decomposition of the metabolic fluxes into separate components. Because of the constraints applied to the metabolic model, the set of demand fluxes ($J_\gamma$ in Eq. (1)) included the demand of each biomass building block, the acetate excretion flux, and the ATP maintenance flux. By applying FDM, we separated the metabolic fluxes into individual components (Supplementary Data 3). Flux components associated with the respiration of glucose to $CO_2$ and aerobic fermentation to $CO_2$ and acetate were obtained by coupling the two flux components associated with ATP maintenance and acetate production, as illustrated before (Fig. 1b, c). The flux components associated with several biomass components were also found to be

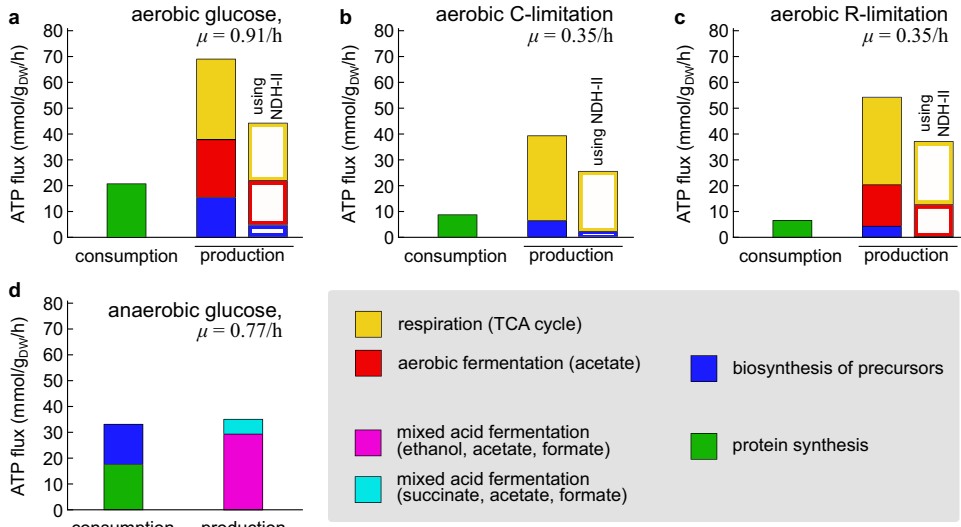

**Fig. 3 | Energetic balance in *E. coli* cells. a** Comparison of the cellular ATP consumption (dominated by protein synthesis, green) and production in aerobic glucose minimal medium. The ATP production is broken down according to the source: respiration (yellow), aerobic fermentation (red) and the synthesis of biomass precursors (blue) sources. Filled bars indicate the estimated ATP production using the FBA solution with maximum yield (using the efficient NADH dehydrogenase I, encoded by the *nuo* genes). Empty bars are obtained in flux solutions using the less efficient NADH dehydrogenase II (*ndh* gene; see text). **b, c** Same as the previous panel, but for slow, C-limited and R-limited growth. **d** The same

breakdown of energy production and consumption was obtained for anaerobically-grown cells. Application of FBA and FDM to this case is analogous to the aerobic case, and is described in Supplementary Fig. 5 and Supplementary Note 2), and allowed us to determine the ATP production flux from two distinct modes of mixed acid fermentation (purple and cyan, see Supplementary Fig. 5g) and the ATP consumption associated with the biosynthesis of biomass precursors (blue). The estimated production and consumption fluxes are close to each other, around 35 mmol ATP/$g_{DW}$/h.

coupled to the ATP maintenance-associated flux component, and were similarly combined (see Supplementary Note 2.3 for details). We illustrate in Fig. 2e and Supplementary Fig. 2a the flux decomposition for some representative reactions from the central carbon pathways across growth rates. As the carbon flows from upper to lower glycolysis and the TCA cycle, precursors are diverted towards biosynthetic pathways, and therefore reactions are increasingly focused toward energy production. Biosynthetic fluxes, e.g. the glutamate dehydrogenase reaction (GDH) are similar between the two growth limitations. The decomposition of the intake fluxes (Fig. 2f, Supplementary Fig. 2b) reveals the fraction of nutrient sources associated with energy production and to the synthesis of biomass precursors. The functional decomposition of the carbon uptake flux mirrors that of upper/mid glycolysis (e.g. enolase, ENO): depending on the condition, 25% to 50% of the carbon is consumed for energy production (~30% in the reference condition), while the remainder is allocated to the biosynthesis of biomass precursors, mostly amino acids (shades of blue). Nitrogen and phosphate are instead used exclusively for the biosynthesis of biomass components, in particular amino acids (nitrogen) and nucleotides (phosphate).

As mentioned above, the flux to the Pentose Phosphate Pathway (PPP) is mostly associated with biomass production, but has a large mixed component $F_i^{mix}$, as shown for the G6P dehydrogenase (G6PDH) reaction. To explore the prevalence of sign-mismatched flux components across the entire network, we computed $F_i^{mix}$ before (Supplementary Fig. 3a–c) and after (Supplementary Fig. 3d–f) the flux coupling procedure. Mixed functional fractions for TCA and oxidative phosphorylation reactions are greatly reduced by the coupling. Instead, large mixed functional components for reactions in the PPP, upper glycolysis and nucleotide metabolism persist after the coupling. We studied the origin of such mixed components by dissecting the synthesis and consumption fluxes of each metabolite in terms of flux components (Supplementary Fig. 3g–i, Supplementary Note 3), and identified key metabolites such as NADPH and DHAP that are produced and

consumed in association with different metabolic functions, thus necessitating anaplerotic reactions with mixed functional assignation.

**Energetic metabolism**

The functional decomposition of the network fluxes allowed us to study in detail the energetic metabolism of the cell. The maintenance ATP hydrolysis flux models the energetic requirements of homeostasis and growth in metabolic models. As a result of our flux coupling procedure, each flux component has an associated net ATP hydrolysis flux, which is balanced through the ATP maintenance reaction. Depending on its sign, we refer to this flux as the ATP/energy production or consumption associated with the flux component; for flux modes, this flux represents the ATP/energy cost or yield associated with the metabolic function.

Protein synthesis (tRNA charging and AA polymerization) is generally thought to be the major energetic burden to the cell[37], at 4 ATP equivalents per residue. The estimated ATP demand is shown as green bar in Fig. 3a–c for reference and slow C- or R-limited growth conditions. We estimated the energy consumption associated with other ATP-consuming processes, namely mRNA turnover and chemotaxis, and we found that these are negligibile compared to protein synthesis (Supplementary Note 4.5). This suggests that the cost of protein synthesis well represents the cost due to the known cellular processes. However, there may be additional sources of ATP consumption, such as metabolic futile cycles, including those originating from membrane leakage, which are not fully understood and remain largely unknown[38].

When comparing the ATP production fluxes to the estimated consumption due to protein synthesis, we found that the total production flux was several-fold higher than the expected consumption flux (compare the filled bars in Fig. 3a–c; see also Supplementary Fig. 4a). Most of the energy production, about 70 mmol ATP/$g_{DW}$/h in the reference condition, is associated with the respiration and fermentation pathways (yellow and red, respectively, yielding ~55 ATP/$g_{DW}$/h combined). In slow, C-limited growth, only the respiration

pathway is active, and the ATP flux is reduced (Fig. 3a). However, in the case of R-limitation, the energetic fluxes appear to be similar to those observed in the reference condition, with a roughly equal share for flux associated with respiration (yellow) and aerobic fermentation (red) (Fig. 3b). This is due to the combination of acetate production (Fig. 2c) and high total ATP production (Fig. 2d) observed for slow-growing R-limited cells.

Counter intuitively, the biosynthesis of biomass building blocks also has a positive net contribution to the cellular energy budget –a fact that we will discuss below. In fact, the ATP produced in conjunction with the biomass building blocks (filled blue bars) is close to the energetic demand associated with protein synthesis (in green). Even under fast-growth conditions, it seems that most of the flux through the respiration and fermentation pathways, which are solely dedicated to ATP synthesis, is not needed to power the known energy-consuming processes.

**Efficiency of the electron transport chain.** While it is possible that the ATP in excess might be consumed in metabolic futile cycles or by other unknown processes, an alternative explanation could be that the metabolic model overestimated the ATP production fluxes. The cell can modulate the efficiency of the electron transport chain (ETC), and thus the overall production of ATP by oxidative phosphorylation, by expressing different NADH dehydrogenase enzymes[39]. FBA fluxes are obtained under the assumption of minimal carbon consumption, which leads to the use of the efficient NDH-I protein complex (*nuo* genes). On the other hand, quantitative proteomics data[9,40] showed that the concentration of NDH-I varied across conditions. Instead, the concentration of the inefficient NADH dehydrogenase II (NDH-II, *ndh* gene) showed trends to those of NDH-I opposite trend across conditions, and became comparable at either fast or slow, R-limited growth (Supplementary Fig. 4b). This led us to hypothesize that the cell might use the inefficient NDH-II enzyme at least in some growth conditions.

When recalculating the FBA fluxes assuming that NDH-II was used instead of NDH-I (i.e. modeling a *nuo⁻* strain), both the energetic yield of reduced electron carriers through the ETC, and the efficiency of both respiration and fermentation pathways were reduced (Supplementary Fig. 4c,d). Consequently, the predicted energy production fluxes were reduced by about 40%, thus partially reconciling the estimated energy production flux with the theoretical cellular demand (Fig. 3a–c, open columns; see also Supplementary Fig. 4e), but still exceeding the energy consumption flux by 2-4 fold.

**Energy balance in anaerobic growth.** If the reduced efficiency of the ETC is indeed responsible for the excess ATP production observed, then we would anticipate a better agreement between produced and consumed ATP under anaerobic conditions, in which ATP is exclusively generated through substrate-level phosphorylation with well-established stoichiometries. To test this hypothesis, we measured the growth rate and main metabolic fluxes of glucose-limited cells in anaerobic growth (see Methods), and used the data to model the intracellular fluxes with FBA (Supplementary Fig. 5a–f). The estimated maintenance ATP flux is much lower compared to the aerobic case, and almost growth-independent at ~ 20 mmol ATP/$g_{DW}$/h. This lower value is also consistent with data from an earlier report[13] in which, however, the difference between aerobic and anaerobic growth was not emphasized.

Application of FDM to anaerobic growth is similar to that of aerobic growth, except the constraint on acetate excretion is substituted by a constraint on succinate production, leading to the presence of two distinct mixed acid fermentation functional modes (Supplementary Fig. 5g). After minimizing the prevalence of mixed functional components across the network (Supplementary Fig. 5h,i), we obtained the functional decomposition for all fluxes in anaerobic conditions (Supplementary Data 4). The total ATP synthesized by the

energetic pathways is about ~ 35 mmol ATP/$g_{DW}$/h in the reference condition (Fig. 3d, cyan and purple), and matches well the predicted consumption by biosynthetic activities (blue and green). This result further suggest that the disagreement between the ATP produced and consumed in aerobic conditions is at least partially caused by the ETC operating with a reduced efficiency.

**Biosynthesis-associated energy flux.** As noted above, the biosynthetic pathways are associated with a net production of energy in aerobic conditions, and a net consumption in anaerobic conditions. Further analysis of the biosynthetic flux components (Supplementary Data 6 and Supplementary Note 4) reveals that, in aerobic conditions, the net energy production derives from the NADH produced by the central carbon pathways as they supply carbon precursors to the biosynthetic pathways (Supplementary Fig. 5j). However, in anaerobic conditions, the cell is unable to convert such NADH to energy via oxidative phosphorylation (Supplementary Fig. 5k). Since substrate-level phosphorylation is unable to generate sufficient ATP to drive the biosynthetic reactions, the ATP requirement of the biosynthetic pathways must be met by the fermentation pathways.

## Global definition of functional modules and costs

We then turned to the analysis of the global structure of the flux and functional decomposition. From this point on, we focused on the flux solutions obtained in aerobic conditions with the high-efficiency NDH-I enzyme. We first considered the functional decomposition in the reference condition, $\mu \approx 1$/h, and performed a hierarchical clustering of the functional shares $F_i^{(y)}$. The analysis, summarized in Fig. 4a and reported in Supplementary Data 5, uncovered a modular structure of the metabolic network: most reactions are highly specialized, and contribute to few biological functions. Groups of functionally similar reactions can be obtained as a function of a threshold on their mutual distance (see Methods), thus allowing us to define a hierarchy of functional modules in the metabolic network of *E. coli*. These functional modules were associated with the synthesis of individual (proline, histidine, arginine, lysine) or groups of amino acids (aromatic AA, alanine/valine/leucine, cysteine/methionine); these modules match well known biosynthetic pathways and superpathways[41]. As expected, TCA cycle reactions are mostly associated with energy production, while other reactions from the central carbon pathways, such as those included in upper/lower glycolysis, electron transport chain and pentose phosphate pathway, have more broadly distributed functions associated with the synthesis of several amino acids and/or energy. In sum, these results indicate that our approach is able to recapitulate the known biological functions of the metabolic network of *E. coli*.

Each flux component, Eq. (2), describes the pathway used by the cell to synthesize either ATP or biomass building blocks, including their associated ATP production or consumption, including contributions from both the synthesis of biosynthetic precursors and due to the flux coupling (Fig. 1b, c and Supplementary Note 2). As an example, the flux mode describing the synthesis of glutamine from glucose is shown in Supplementary Fig. 6. Using the flux components, we computed carbon and energy costs for each biomass component and, in particular, for amino acids and nucleotides. These are shown in Fig. 4b, c in either costs per molecule (Fig. 4b) or per amount of cellular biomass (Fig. 4c) in glucose minimal medium, and reported in Supplementary Data 8. We observed that the overall ATP balance is positive for many amino acids, especially for the most represented in the biomass composition (glycine, leucine). On the other hand, nucleotide biosynthesis is expensive in terms of both energy and carbon consumption.

We validated these results in two ways. Firstly, the energetic costs associated with the biosynthetic pathways of each amino acid have been well characterized[31]. These costs are computed for each pathway starting from carbon precursors, rather than from the external carbon

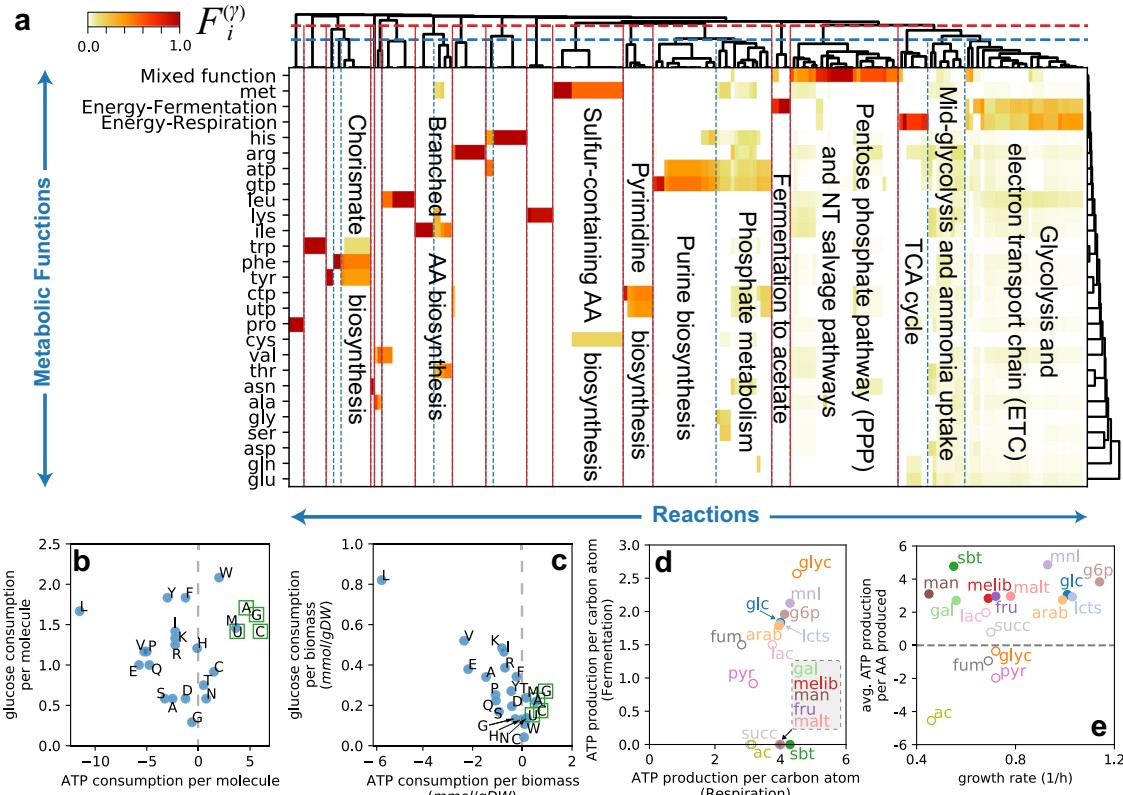

**Fig. 4 | Metabolic costs for biosynthetic activities. a** Hierarchical clustering of the functional decomposition $F_i^{(\gamma)}$, restricted to the production of energy, amino acids and nucleotides, in the reference condition (214 reactions). Clustering reactions based on the similarity of the functional profiles allows to define a hierarchy of functional modules. Here we show the modules arising from two different choices of the similarity threshold (horizontal lines in the top dendrogram); the resulting functional modules are summarized in Supplementary Data 5. **b** Carbon and energy (ATP) costs associated with the biosynthesis of amino acids (blue circles) and nucleotides (green squares), in glucose media. Nucleotides are more expensive than most amino acids, both in terms of carbon substrate and energy. Instead, the biosynthesis of several amino acids is associated with the net production of energy (i.e. a negative cost). Such coupling emerges from flux-balancing the pathway intermediates (since the flux components $\xi^{(\gamma)}$ are flux-balanced) and from the removal of sign-mismatched flux components (see Supplementary Note 2). **c** Same as before, but weighting the costs by the demand of the building blocks (in mmol per gram of dry weight). Leucine production is associated with a large glucose

source. In order to compare to these costs, we removed from the flux components all reactions belonging to the central carbon pathways, and computed the energetic balance for the biosynthetic reactions only. Reassuringly, the resulting energy costs were positive and matched well those reported in ref. 31 (Supplementary Fig. 7a). As a second test, we compared our carbon and ATP costs to those obtained in a previous analysis from ref. 32. In this study, the costs were computed either by manual counting or through optimization on a simplified *E. coli* network. The comparison displayed a general agreement for most amino acids, see Supplementary Fig. 7b,c. However, our approach led to much smaller ATP costs for the biosynthesis of a few amino acids compared to ref. 32. As illustrated in Supplementary Fig. 7d, this was traced back to the different assumptions involved in the methods: while NADPH requirements were simply converted to energetic costs in ref. 32, our methods enforces the complete flux balance of NADPH, leading to significant differences for some amino acids.

By using experimentally determined acetate production fluxes from ref. 7, as well as the biomass composition for C-limited cells (Fig. 2a), we were able to repeat the analysis for cells grown aerobically

uptake, but also to a large ATP synthesis flux. **d** Scatter plot of respiration and fermentation ATP yields per intaken carbon atom, for a variety of carbon sources (Supplementary Data S5 and S7). The yields were calculated by repeating the FBA and FDM calculations for cells growing on different substrates, constraining the growth rate and fermentation fluxes to the experimental data from ref. 7, and further assuming the same ATP maintenance flux parameters and growth-dependent biomass composition as for carbon-limited growth. Open symbols indicate the yields for the glycolytic carbon sources, while filled symbols indicate the yields for gluconeogenic substrates. An ATP yield of 0 is shown for the fermentation pathway if cells do not produce acetate in the corresponding condition. **e** The average net ATP production per AA (weighted by AA abundance) for different carbon substrates, as a function of the corresponding growth rate (Supplementary Data S5 and S7). For most glycolytic carbon sources, 3 to 5 ATP molecules are regenerated for each amino acid synthesized. Instead, the net energy balance is close to zero, or negative (net consumption), in cells growing on gluconeogenic carbon sources (open circles), and acetate in particular.

on a variety of carbon sources. We found that the efficiency of both the energetic and biosynthetic pathways depended strongly on the substrate, as seen in Fig. 4d. The energy yields of the respiration pathways (Fig. 4d) are ~ 4 ATP per C atom for glycolytic carbon sources (open circles), and 2.5 to 3 ATP/C atom for gluconeogenic carbon sources such as acetate, succinate, fumarate and pyruvate (filled symbols). The energetic efficiency of the fermentation pathway is typically 2 ATP/C atom for glycolytic carbon sources, while gluconeogenic carbon sources yield between 1 ATP/C atom (pyruvate) to ~ 2.6 ATP/C atom (glycerol).

The net energy yield associated with amino acid biosynthesis is also strongly affected by the carbon substrate on which the cells are grown (Fig. 4e). Amino acid biosynthesis is on average much more expensive for cells grown on gluconeogenic carbon sources (empty symbols), compared to glycolytic substrates. For example, the synthesis of an amino acid on mannose produces about 3 ATP, while it consumes more than 4 ATP's when cells are grown on acetate. (See Supplementary Fig. 7e-l for a breakdown of the costs for individual amino acids and nucleotides.)

Interestingly, no particular differences among glycolytic substrates is seen in either the energetic efficiencies (Fig. 4d, filled circles) or the average ATP produced per synthesized amino acid (Fig. 4e), in contrast to the wide range of growth rates achieved by cells reared on these substrates (shown as x-axis in Fig. 4e). For example, owing to their similar chemical composition, mannose and glucose have very similar energetic yields, but the growth rates for cells grown on the two substrates differ by a factor 3. Therefore, the growth rates achieved with different glycolytic substrates do not appear to be determined by differences in the energetic parameters of each carbon source.

The comparison between aerobic and anaerobic growth also suggests that cellular energetics do not impact cellular growth on glycolytic sources significantly. In anaerobic conditions, the glucose intake flux for wild-type *E. coli* is > 2 times than that in aerobic conditions, with a corresponding reduction in growth yield (Supplementary Fig. 5l). The increase in overall glucose flux is mainly driven by the ~ 7-fold increase in energy-associated glucose intake required for aerobic growth (~ 2 mmol/$g_{DW}$/h, Fig. 2f) compared to anaerobic growth (~ 15 mmol/$g_{DW}$/h, Supplementary Fig. 5m) at the same growth rate of ~ 0.75/h. This increase is necessary to compensate for the drop in energetic efficiency in anaerobic conditions, yielding at most 2.5 ATP per glucose molecule (Supplementary Fig. 5g), compared to up to 24 ATPs aerobically (Supplementary Fig. 4d). Despite these large changes in growth yield and energetic efficiency, the maximum growth rate on glucose is only reduced by less than 20%. Overall, our results suggest that neither carbon availability nor energy is the main limiting factors for aerobic growth on glycolytic carbon sources. Rather, the differences in growth rates might stem from the varied expression levels of the catabolic proteins[9,42] and are possibly modulated by the expression of other underutilized proteins[28].

## Functional decomposition of the proteome

The functional decomposition described above defined functional shares for each flux-carrying metabolic reaction in the cell. In turn, these functional shares can be used to generate a functional decomposition for the corresponding metabolic proteins. To do so, we made use of highly accurate experimental protein abundances[9,40] obtained for cells grown in conditions matching those explored above, namely carbon and translational limitation. We obtained protein abundances for a total of 2017 out of 4312 protein-coding genes in *E. coli* (Fig. 5a), expressed as fraction of the total protein mass. Protein mass fractions, a direct output of mass-spectrometric analysis pipelines, are a convenient measure of absolute protein abundance[9] approximately proportional to protein concentrations (because of the constancy of the total concentration of proteins across conditions in *E. coli*[3,43]). After computing the functional decomposition in each growth condition, we made use of the gene-protein-reaction matrix of the iML1515 model to associate the reactions to expressed proteins. Overall, 412 proteins were both associated with flux-carrying reactions and detected in the reference condition; the numbers are similar for the other growth conditions. For these reactions, the joint use of the experimental protein abundances and of the functional decomposition allowed us to quantify the contribution of each enzyme to the various metabolic functions, as illustrated in Fig. 5b for the enzyme enolase. For enzymes catalyzing multiple reactions (including various promiscuous biosynthetic enzymes such as ArgD, AspC and Ndk), we took the average functional shares of each reaction, weighted by the flux magnitudes (see Supplementary Note 5 for details).

## Protein costs associated with energy production

The protein fraction associated with energy production in carbon-limited growth is similar to previous estimates[7,23], increasing from about 8% to 12% of the total proteome (Fig. 5c) and switching from a mix of respiration and aerobic fermentation at fast growth, to respiration only at slow growth. Given the decreased energetic flux in slow, carbon-limited conditions (Fig. 3a, b and Supplementary Fig. 4a), the overall energetic efficiency of the cell (ATP flux per unit of invested proteins) decreases at slow growth (Supplementary Fig. 8a, red circles). The observed patterns are quite different in R-limited cells: the proteome shares allocated to energy production via either respiration or fermentation are mostly independent on the growth rate (Fig. 5d), mirroring the lack of change observed for the energy flux (Fig. 3c). As a consequence, the efficiency of the energetic pathways is mostly constant across growth rates (Supplementary Fig. 8a, green squares). The protein efficiency for the respiration and fermentation pathways can be obtained by comparing the associated protein shares to the corresponding ATP fluxes. Respiration pathway requires about twice the proteins associated with aerobic fermentation (Fig. 5e), consistently with previous analyses[7,23].

## Protein costs associated with biomass production

Globally, the protein fraction associated with biosynthetic activities (Fig. 5f, g) ranges between 20 and 30%, and decreases in slow growth conditions. The protein cost is dominated by amino acid synthesis (shades of blue). The biosynthesis of the other components (RNA, DNA, lipids, cofactors) only makes use of a small fraction of the total proteome, less than 10% in total. In the reference condition, the synthesis of methionine was by far the most expensive process, requiring more than 6% of the total proteome mass (red bar in Fig. 5h). Most of this cost is due to the highly inefficient enzyme homocysteine methyltransferase (MetE). Instead, the relatively small production costs of proline, glutamine and glutamate were the only ones dominated by the shared central carbon pathways (Supplementary Fig. 9a).

We looked at the relationships between enzyme abundances and demand fluxes for individual amino acids. Across conditions, the protein mass fractions associated with each amino acid and nucleotide scaled remarkably linearly with the corresponding demand flux, with more proteins being allocated in presence of higher fluxes (Fig. 5i, Supplementary Fig. 10a–d). These results are consistent with linear relations observed for transcriptional reporters[5] or for protein sectors[6,9] in *E. coli*, and more recently in yeast[44]. The slopes and y-intercepts (offsets) of the linear relations (Supplementary Data 8), are important parameters that reflect the overall efficiency of the pathway[6,27] and the capacity of the pathway to rapidly change flux in dynamic conditions[40,45], respectively. The slopes and offsets are summarized in Fig. 5j, k. Overall, the protein offsets were similar (at most a 25-fold difference, with 16 out of 20 in the range 0.15% to 0.35%) for most amino acids, while slopes varied over a broader range (more than two orders of magnitude). Methionine (M) biosynthesis (inset in Fig. 5i) is by far the most expensive process across all conditions, and had the steepest slope among all amino acids (Fig. 5j). Other expensive processes are the biosynthesis of tryptophan (W) and histidine (H), as indicated by their large slopes; intriguingly, these three amino acids were also reported to be the most expensive in yeast[46]. On the other hand, leucine (L) and arginine (R) stand out as the amino acids whose associated proteome had the largest offsets (Fig. 5k), while having only moderate slopes in C-limitation. In contrast, the biosynthetic enzymes are expressed at very low levels in rich media (Fig. 5l). This suggests that glucose limitation specifically impacts the expression of biosynthetic enzymes for arginine and leucine, possibly indicating that their biosynthesis becomes growth-limiting in slow, carbon-limited growth.

Overall, results for R-limited growth were similar to those obtained for C-limitation (Supplementary Fig. 10a–d), with the notable exception of methionine which presented a much larger protein offset (Supplementary Fig. 10e, f). Protein costs for the synthesis of purines was 3 to 4 times that of pyrimidine (Supplementary Fig. 10b,d), and larger than most amino acids other than methionine, although the overall allocation towards nucleotide synthesis is small compared to

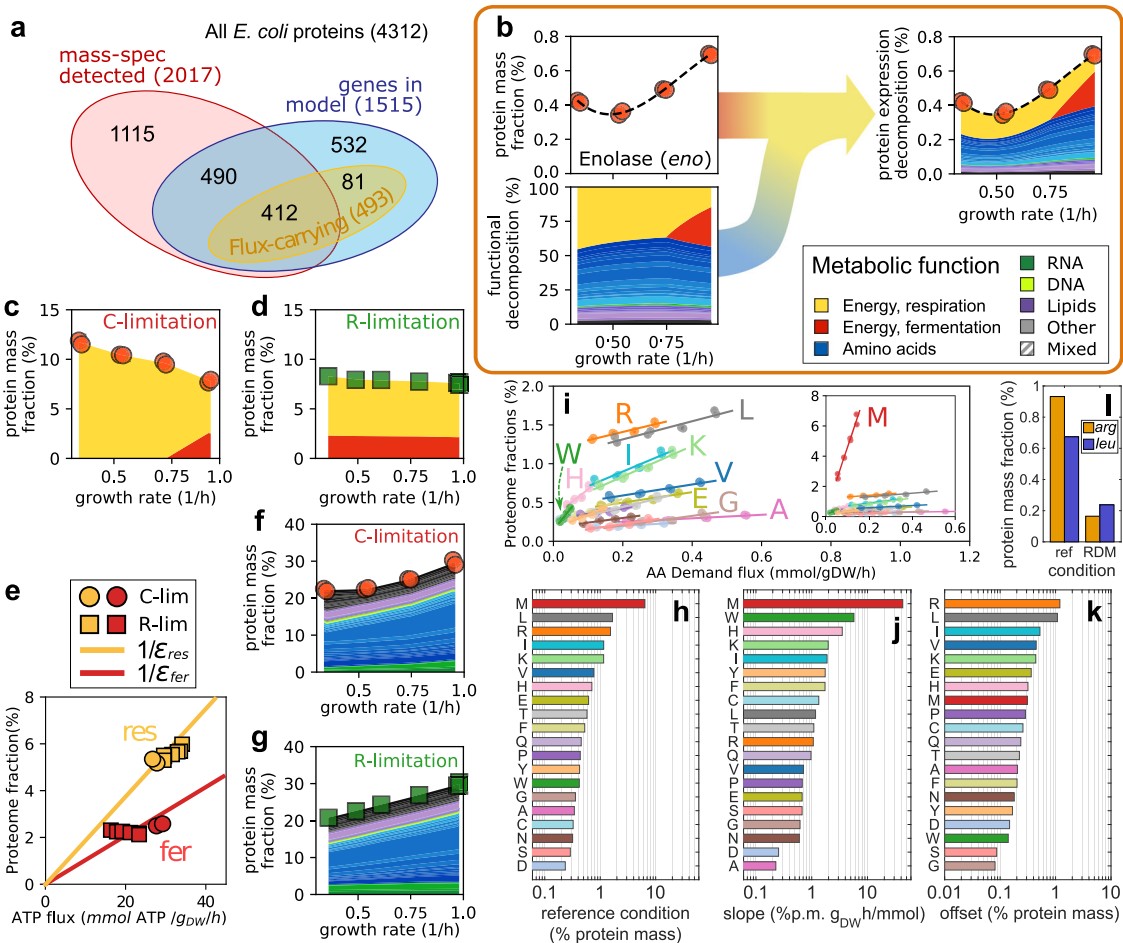

**Fig. 5 | Cellular allocation of metabolic proteins across conditions. a** Overlap between metabolic proteins (included in the iML1515 metabolic model), proteins associated with active reactions in the reference condition, and proteins detected in the same condition[40]. **b** Experimental protein abundances can be matched with the reaction functional decomposition to obtain a functional decomposition for metabolic proteins. In this example we combine protein mass fractions for the Enolase enzyme in glucose-limited cells (red dots, from Ref. [40]) with the functional decomposition in the same conditions, allowing us to define function-specific protein shares (colored bands; see legend on the right). **c** Total abundance of energy-associated proteome $\phi_E$ in C-limited growth, obtained by summing over the energy-associated protein shares of all metabolic reactions. Colors indicate respiration ($\phi_{E,r}$, yellow) and fermentation ($\phi_{E,f}$, red). **d** Same as (**c**), but for R-limited conditions (protein data from Ref. [9]). **e** Protein cost of the respiratory (yellow) and fermentative (red) pathways. Solid lines are best fits lines passing through the origin; the slopes indicate the inverse of the protein efficiencies of the two pathways. We obtained $\varepsilon_{res} = 5.33 \pm 0.15$ and $\varepsilon_{fer} = 9.62 \pm 0.37$ (mean ± standard error), in units of ATP flux (mmol/$g_{DW}$/h) per percent of allocated proteome. **f** Total proteome mass fraction associated with the biosynthesis of biomass building

blocks in C-limited conditions, obtained in the same way the energy-associated proteome fraction was computed in (**c**). Colors indicate individual building blocks. **g** Same as (**f**), but for R-limited conditions. **h** Protein mass fractions associated with each amino acid in the reference condition. **i** As the growth rate is varied in C-limited conditions, the protein shares allocated to the biosynthesis of each amino acid (symbols) are well described by linear function of the demand flux of the amino acid (solid lines indicate best fits). The inset shows the protein mass fraction associated with the biosynthesis of methionine, which is dominated by MetE. Letters indicate a few amino acids; colors match those used in the bars in panel (h). Similar linear relations are also observed in R-limitation (Supplementary Fig. 10c); the values of slopes and offsets (y-intercepts) for both limitation series are reported in Supplementary Data 8. **j,k** Slopes and offsets (y-intercepts) of the linear relations shown in panel (i). **l** Protein mass fractions in the reference condition and in rich defined medium (from Ref. [40]) of the biosynthetic enzymes for arginine and leucine (*arg* and *leu* genes, respectively). Levels in rich media are much lower than in glucose minimal media, as opposed to the lack of change observed in C-limitation for these two amino acids, see panel (i).

the amino acid (Fig. 5f,g, compare green to blue), due to the much smaller number of nucleotides compared to amino acids. We observed that for most amino acids, the slopes of the protein-flux relationships were slightly more steep in R-limitation compared to C-limitation, while the opposite was true for nucleotides. Such small systematic patterns were mostly due to changes in the flux through the central carbon pathways (glycolysis and TCA cycle), which influence how the corresponding proteins are allocated (Supplementary Fig. 9b–d). Slopes were tightly correlated to the ratio of allocated protein and demand flux in the reference condition, related to the so-called effective turnover rate of the pathway, and presented weaker correlations with other quantities such as the carbon costs and demand flux of each amino acid (Supplementary Fig. 11).

## Coarse graining of the proteome according to metabolic function

The functional decomposition described above defined functional shares for each flux-carrying metabolic reaction in the cell. As summarized in Fig. 6a, this corresponds to about 40% of the proteome in the reference condition, of which 7% is associated with energy production (energy protein sector) and 33% to the biosynthesis of biomass precursor (biomass sector). About half of the proteome is not associated with the metabolic model, and an additional 10% is associated, but the corresponding reactions carry no flux. We associated these proteins to specific cellular functions via an iterative GO-terms based categorization. By sequentially selecting the GO-terms associated with the largest protein shares, we were able to categorize > 90% of the

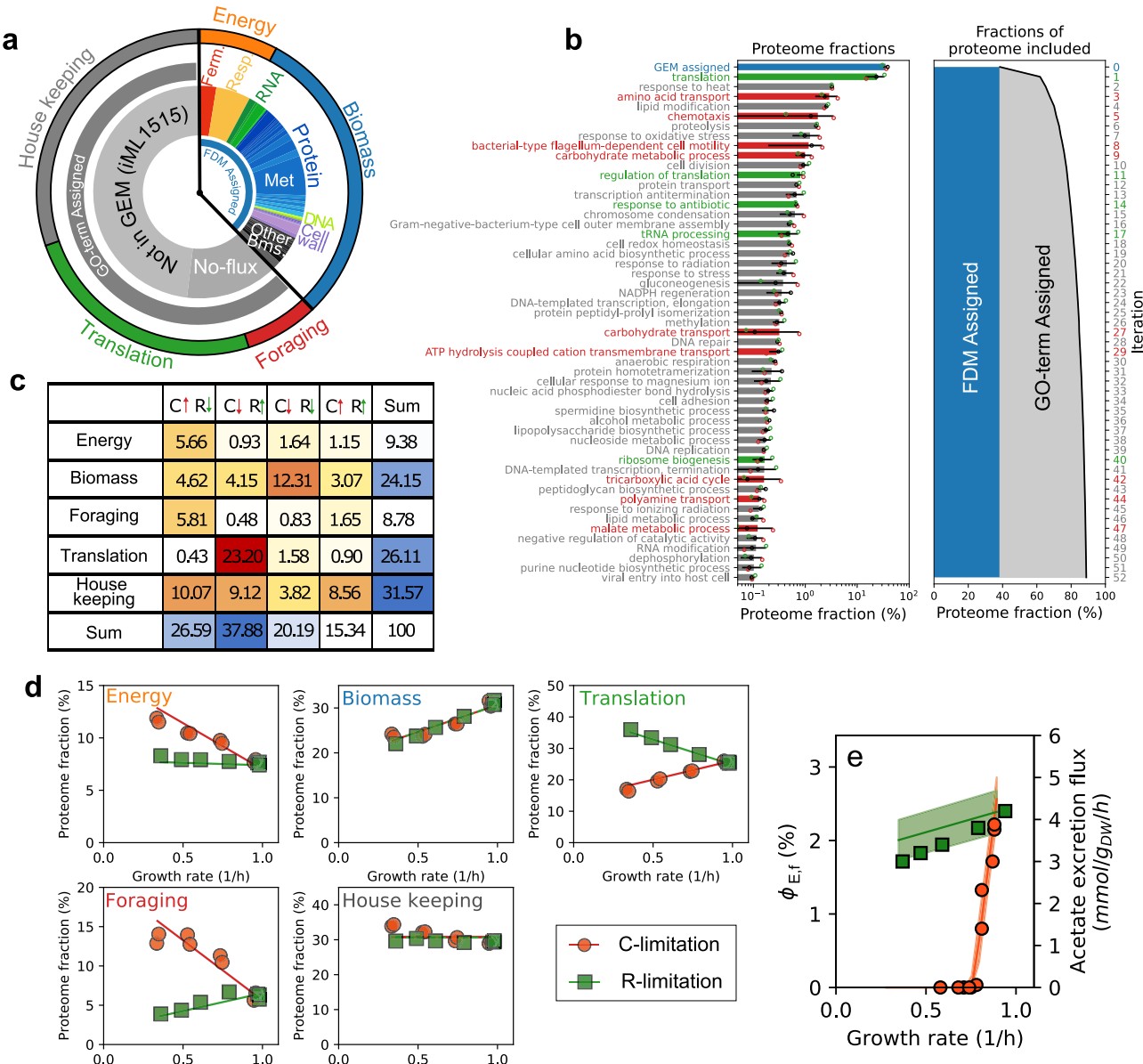

**Fig. 6 | Function-based global classification and coarse-graining of the expressed proteome. a** Summary of the functional decomposition of *E. coli* proteome in glucose minimal medium. The fraction of the proteome contributing to a defined metabolic function (GEM-assigned) is about 40%, with proteins involved in the biosynthesis of cellular building blocks ("biomass" sector) about 32%, and energy-producing proteins (energy sector) close to 8%. The remainder of the proteome is divided between non-metabolic proteins (e.g. ribosomal or motility genes), and proteins associated with zero-flux reactions (e.g. nutrient transporters). Using a GO-term enrichment analysis (see next panel) these were assigned to three translation, foraging (motility and nutrient assimilation) and house-keeping protein sectors. **b** An iterative GO-term assignment procedure (see Methods) was used to assign specific biological functions to non-metabolic proteins. At each steps, the GO-term associated with the largest proteome share is selected, and the corresponding genes are assigned exclusively to that GO-term. Bars and error bars indicate mean ± SD (*n* = 3, data shown as dots). Colors indicate the three coarse-grained sectors indicated in the previous panel. **c** For each protein in the five functional sectors, we summarized changes in protein levels with a binary

classification[6] describing whether the protein is up- or down-regulated in C- or R-limitation, as indicated by the arrows. For all sectors except the "housekeeping", most of the proteome (here corresponding to the reference condition) is associated with only one of the four possible combinations, indicating that most proteins are consistently regulated across conditions. **d** Protein mass fractions associated with the five functional sectors across growth rates for C-limited (red circles) and R-limited (green squares) growth. Solid lines indicate the best fit for a coarse-grained model of protein allocation. **e** Using the calculated energy production fluxes and protein efficiencies for the respiration and fermentation pathways, as well as the modeled total protein abundance, the coarse grained model predicts the partitioning between respiration and fermentation-associated proteins (see Supplementary Note 6). Here we show the predicted allocation towards fermentation proteins and the associated acetate excretion flux (the solid lines and bands indicate mean and 68% confidence bands, obtained by propagating mean and SEM for the efficiencies), as well the experimentally determined acetate excretion fluxes (same symbols as in the previous panel), for C-limited and R-limited growth.

proteome mass using ~ 50 GO-terms, Fig. 6b. In order to simplify the description of the proteome, we grouped all translation-associated GO-terms into a translation sector; similarly, GO-terms associated with nutrient transport and catabolism, as well as motility, were grouped

into a single foraging protein sector; the remainder of the GO-terms were grouped into a housekeeping sector. Together with the energy and biomass metabolic sectors, these define a five-way function-based classification of the proteome.

We first assessed whether proteins within the same functional sector are similarly regulated across conditions. Each protein was previously classified depending on their change in protein abundance upon carbon starvation or translation limitation[9]. Hence, we were able to break down each functional sector into four different components, corresponding to the four possible combinations of up- and down-regulation in the two growth limitations (Fig. 6c; see also Supplementary Fig. 12a–c). For most functional groups, proteins appear to be consistently regulated across conditions, with the vast majority of the associated proteome belonging to only one of the four regulatory groups. This is most evident for the translation functional group, where the vast majority of proteins are similarly upregulated in translation-limited conditions and downregulated in carbon-limiting conditions. Most proteins in the energy and foraging groups show the opposite behavior, i.e. are upregulated in carbon-limited conditions and downregulated in translation-limited conditions. Proteins in the biomass sector are downregulated in both growth limitations. Finally, housekeeping proteins are regulated more heterogeneously. This was also expected given the wide variety of GO-terms associated with the proteins in this functional group. Similar results were obtained by considering a more detailed binary classification based on three (rather than two) different growth limitations (Supplementary Fig. 12d–g). We then computed the protein abundance of the functional sectors as a whole, which can be seen plotted against the growth rate in Fig. 6d. The growth-dependence of the sectors recapitulates what was seen at the level of individual proteins in Fig. 6c: The energy and foraging sectors increase strongly in carbon-limited conditions, while the translation sector is upregulated in R-limited conditions. Allocation towards the biomass sector is proportional to the growth rate, while the housekeeping sector changes little across conditions.

The trends observed for the protein sectors, as well as the observed switch between respiration and fermentation, can be captured quantitatively by a single phenomenological model of protein allocation. The protein mass of each sector changes according to two model parameters, the quality of the carbon source $\nu_C$ and the translational capacity of the cell $\nu_R$. These two parameters determine the cellular phenotype[2]: changes in either parameter leads to simultaneous changes in both the cellular growth rate $\mu$ and in the size of the protein sectors. The expression of the foraging (including a variety of transporters and chemotactic proteins) and translation sectors respond specifically to changes in the carbon quality and in the translation capacity, respectively. Expression of the biomass sector is linearly related to growth rate, while the housekeeping sector is constant. Finally, we assumed a regulatory constraint setting the energy sector in response to carbon starvation and translational limitation of the following form:

$$\phi_E = \phi_{E,0} + \kappa_C \frac{\mu}{\nu_C} + \kappa_R \frac{\mu}{\nu_R}, \qquad (4)$$

where $\kappa_C$ and $\kappa_R$ determine change in protein allocation in response to the specific growth limitation. As detailed in Supplementary Note 6.3, these two terms arise from the reduced efficiency of the energetic pathways in carbon-limited conditions (Supplementary Fig. 8a), and from increased energetic demands per unit of biomass in translational-limited conditions (Supplementary Fig. 8b). The fitted model (solid lines in Fig. 6d) correctly recapitulates the experimental data. Because the model explicitly accounts for the protein share associated with energy production, it can also account for the impact of protein allocation on acetate overflow. Assuming that the respiration and fermentation-associated components of $\phi_E$ generate an ATP flux in proportion to their proteome share (using the efficiencies determined in Fig. 5e). Without changing any of the parameters leading to the fit in Fig. 6d, the model predicts a sharp decrease in the protein share associated with fermentation $\phi_{E,f}$ in carbon-limited growth, and sustained fermentation in R-limited growth (Fig. 6e, solid lines), in

agreement with the experimental acetate excretion fluxes (Fig. 6e, symbols). Thus, the model is able to simultaneously capture the observed protein allocation patterns and the metabolic switch between respiration and fermentation.

## Discussion

In this work, we presented a computational method termed Functional Decomposition of Metabolism (FDM) for studying cellular metabolism and protein allocation based on the decomposition of metabolic fluxes into distinct functional components. This framework allowed us to comprehensively evaluate metabolic costs and protein burdens associated with each metabolic function, a feat impossible to achieve from the analysis of single reactions or protein abundances because of the deeply interconnected nature of metabolic networks.

The flux decomposition at the core of FDM is fundamentally derived from mathematical properties of FBA solutions. Furthermore, it does not rely on parameters such as kinetic constants, nor requires simplifying hypothesis on the structure of the metabolic network: FDM can be generally applied to any network, as long as the application of FBA (or other optimization approaches) estimates correctly the intracellular fluxes. At the core of FDM is a quantitative definition of system-level metabolic functions for each reaction and protein. Each of these metabolic functions can be associated with multiple activities, e.g. synthesis of amino acid and ATP production, when these are tightly coupled. Furthermore, flux constraints are often applied to the network in order to improve the agreement with experimentally determined fluxes; these constraints are fully accounted for by the flux decomposition, at the cost of introducing associated metabolic functions, as shown in Fig. 1b for the constraint on acetate production. However, the interpretation of such flux components might be not straightforward due to sign-mismatch issues which arise when, for a given reaction, different flux components have opposite signs. These flux patterns are expected to arise generically even in the absence of external constraints, for example when the synthesis of a biomass component is coupled to the net production of energy. Coupling flux components by combining different metabolic functions is an effective strategy to make their biological interpretation more direct and reduce the sign-mismatch prevalence across the metabolic network (as shown in Fig. 1c for the case of acetate). However, when applying FDM to other systems, the choice of which processes to couple depends on the network, optimization process, and constraints at hand, and must be evaluated on a case-by-case basis. In this work, we observed that minimizing the prevalence of sign-mismatch also provided the most transparent and intuitive set of metabolic functions. We thus propose that this approach could guide the application of FDM in other organisms.

One of the most immediate results enabled by FDM was the calculation of specific ATP and carbon costs for the biosynthesis of individual biomass component, as well as the ATP yields of the energetic pathways. These were generally uncorrelated with the rate of aerobic growth, although we noticed that gluconeogenic carbon sources tended to yield less energy and allow for slower growth rates compared to the best glycolytic carbon sources (compare e.g. succinate vs glucose in Fig. 4d, e), or similar to the worst glycolytic sources (e.g. acetate and mannose, both yielding growth rates close to 0.3/h). This suggests that the energetic metabolism might play a more significant role in gluconeogenic growth than in growth on glycolytic substrates. In fact, the determination of carbon and energetic fluxes associated with the biosynthesis of cellular components showed that, for aerobic growth on glucose, the ATP flux produced as by-product of biosynthesis on glucose media is already sufficient to supply for the need for amino acid polymerization, the single biggest ATP expenditure for growing cells (Fig. 3a–c). These results raise fundamental questions on the metabolic purposes of energy biogenesis by respiration and fermentation, which comprise 30% or more of the total carbon flux during aerobic growth on glucose, and are apparently not

needed. Our analysis suggests two possibilities: first, the cell might be inefficiently producing energy via respiration, so that the actual energetic flux is lower than what could be estimated from the observed exchange fluxes. Second, the excess energy might be consumed by unknown processes, including puzzling wasteful energy-spilling pathways[47,48].

We suspect that both possibilities might hold true in the conditions studied here. The opposite changes observed in the levels of the two NADH dehydrogenases NDH-I and NDH-II across conditions (Supplementary Fig. 4b) suggest that *E. coli* cells are able to decouple electron transport from ATP production, thus reducing the production of ATP and bringing it closer (but not equal) to the estimated consumption. In anaerobic conditions, when the electron transport chain is not used, the predicted ATP production matches the costs (Fig. 3d). These results suggest that the energetic pathways in fast-growing *E. coli* cells in aerobic conditions operate with and efficiency far from the maximum allowed by the biochemical constraints. For cells grown in R-limited conditions, we observed high, constant energy production fluxes for both the respiration and fermentation pathways, despite a predicted reduction in ATP demand at slow growth. Such constant flux is accompanied by a constant share of proteome allocated to energy production, irrespective of the growth rate. This might suggest the presence of additional energy-consuming processes in slow, R-limited growth, e.g. additional ribosome turnover due to biogenesis defects[49]. Alternatively, the flux of carbon substrate towards energy production might be set by the carbon availability (which is constant in R-limited conditions), while being independent on the actual energetic demand; the latter could be matched instead by modulating the ATP yield of the energetic pathways, in agreement with the analysis discussed above.

Combining FDM with proteomics data allowed us to calculate the proteome costs associated with the de novo biosynthesis of each cellular component, including not just the contributions from the curated pathways, but also the prorated cost of carbon/nitrogen uptake and energy biogenesis needed for biosynthesis. For the biosynthesis of amino acids and nucleotides, we found the total abundance of the allocated proteins to be linearly increasing functions of the growth rate under carbon catabolic limitation. In addition to the well-known large cost of methionine biosynthesis, we found large protein reserves at slow growth for the production of several amino acids, particularly leucine and arginine. These large protein reserves might indicate that the synthesis of these two amino acids becomes growth-limiting in poor carbon conditions.

The whole-proteome, function-based model of protein allocation enabled by FDM is a step forward in the quantitative modeling of bacterial protein allocation. Our work allowed us reconcile two distinct classes of protein allocation models. The first class includes models based on regulation-based protein sectors[6,9]. In these models, protein sectors are defined based on protein expression patterns, but they do not always correspond to unique biological functions. Models in the second class are formulated with function-based sectors[7,23], and have a narrower scope (e.g. focusing on energetic metabolism). The design of a systematic procedure to functionally classify all of the expressed proteome across conditions (Fig. 6a, b), and the finding that most proteins within each condition-dependent protein sector were consistently regulated across conditions, enabled us to build a quantitative model of protein allocation bridging the two model classes.

Still, the regulation of the energy sector in poor carbon sources is only accounted for by an effective constraint, and linking its share to the underlying regulatory processes is an open problem. The increase in the protein share assigned to energy production might be a regulatory strategy to more efficiently divert flux from biosynthetic activities, or to prepare for a switch to gluconeogenic substrates[50]. In either case, fully explaining the observed patterns likely requires including information on the concentrations of metabolites and the kinetics of the respiration pathway[51,52].

FDM's versatility as a method for analyzing cellular metabolism and protein allocation opens up a wide range of possibilities for future research in bioengineering and systems biology. The systematic evaluation of yields and costs for the production of individual metabolites (Supplementary Data S3 and S8) enabled by FDM has natural applications to the study of microbial cell factories in which the production of a metabolite of interest is maximized. FDM also allowed us to uncover a hierarchy of functional modules in cellular metabolism without supervised knowledge on curated metabolic pathways. Thus, FDM could facilitate the rational design of heterologous metabolic pathways with varying degrees of coupling to other metabolic activities[53]. Consistently with the modularity exhibited by bacterial metabolic networks[54,55], the functional patterns displayed a rich structure (Fig. 3a), with clusters of reactions with similar metabolic functions providing the quantitative counterpart of known biochemical pathways. Thus, the definition of functional shares for each reaction represents a simple alternative to other system-level approaches to the analysis of metabolic function, e.g. based on the exhaustive enumeration of extreme pathways or elementary modes[56].

The simplicity of FBA allows the method to be generally applicable on a variety of organisms and conditions, while the ability to integrate a wide variety of physiological, flux, and protein data using genome-scale metabolic models (as illustrated in Supplementary Fig. 13) provides a general avenue to the analysis of complex multi-omics datasets. Other studies[46,57,58] quantified protein costs using marginal costs within more complex models that require effective enzyme kinetic parameters, which are difficult to determine[26] and can largely affect the flux solutions[25]. Not only our approach is based on a simple (vanilla) FBA framework, but it also provides a concrete interpretation of marginal costs as flux modes. The complete decomposition of fluxes into the complete set of components, Eq. (1) and (2), enables the global functional analysis and permits the application of the flux-coupling procedure, which is necessary to obtain realistic pathways from the original flux modes (as demonstrated in Fig. 1b for the case of acetate excretion). On the other hand, adapting FDM to more complex frameworks, including ME-models[20], RBA[59] and other approaches dependent on enzyme parameters[27,60,61] could provide additional information on cellular fluxes and proteome utilization for processes not captured by simple GEM models. In sum, our findings demonstrates the power of FDM as a framework for analyzing cellular metabolism and protein allocation, and its potential to advance our understanding of metabolic networks in a range of contexts.

## Methods

### Experimental methods

**Bacterial strains.** All strains used in this work are derived from *Escherichia coli* K-12 NCM3722[62–64] and listed in Supplementary Table S1.

**Growth media.** The growth media for aerobic growth is the MOPS-buffered minimal medium from ref. 7. The phosphate-based growth media used for anaerobic growth and for other control samples in aerobic conditions contained 10 mM glucose, 80 mM $K_2HPO_4$, 20 mM $KH_2PO_4$, 10 mM NaCl, 10 mM $NH_4Cl$, 0.5 mM $Na_2SO_4$, a phosphate buffer and a 1000x micronutrient solution. The 1000x micronutrient solution contained 20 mM $FeSO_4$, 500 mM $MgCl_2$, 1 mM $MnCl_2.4H_2O$, 1 mM $CoCl_2.6H_2O$, 1 mM $ZnSO_4.7H_2O$, 1 mM $H_{24}Mo_7N_6O_{24}.4H_2O$, 1 mM $NiSO_4.6H_2O$, 1 mM $CuSO_4.5H_2O$, 1 mM $SeO_2$, 1 mM $H_3BO_4$, 1 mM $CaCl_2$, and 1 mM $MgCl_2$ dissolved in a 0.1 M HCl solution. Carbon limitation was implemented by titrating 3-methyl-benzylalcohol (3MBA) concentration in strains NQ1243, NQ1448, and NQ1554 as well as by growing strain NQ1261 (*ptsG* deletion) in glucose. Translational limitation was obtained by adding sublethal doses of chloramphenicol. Concentrations of nutrients, 3MBA and chloramphenicol in each experiment are reported in Supplementary Data S2.

**Growth measurements.** Growth measurements for aerobic culture were performed as in ref. 5. Briefly, exponential cell growth was performed in a 37 °C water bath shaker at 240 rpm. Cultures were grown in the following three steps: seed culture, pre-culture, and experimental culture. Cells were first grown as seed cultures in LB broth for several hours, then as pre-cultures overnight in an identical medium to the experimental culture. Experimental cultures were started by diluting the exponentially growing pre-culture to an optical density at wavelength 600 nm (OD$_{600}$) of ~ 0.01–0.02. Growth rates were calculated from at least seven OD$_{600}$ points within a range of OD$_{600}$ of ~ 0.04–0.4.

Anaerobic growth was performed similarly to aerobic growth with a few exceptions. All transfers were performed with disposable syringes to avoid oxygen contamination. Aerobic seed cultures were diluted into Hungate tubes for preculture. After overnight growth, the precultures were diluted into fresh Hungate tubes for experimental culture. To avoid atmospheric exposure from removing samples, OD measurements were performed with a Thermo Genesys 20 modified to hold Hungate tubes in place of cuvettes. The culture temperature was kept stable during OD measurements by removing and replacing the Hungate tubes from the water bath shaker within 30 seconds. The OD$_{600}$ measured through the Hungate tubes was equivalent to the OD$_{600}$ measured through a cuvette for the range of 0.04–0.5.

**Metabolite measurements.** Metabolites were prepared and quantified as in ref. 65. Four samples of 200 $\mu$L were pipetted from culture tubes at regularly spaced ODs during exponential growth. For anoxically grown cultures, samples were removed with tuberculin syringes inserted into the rubber stopper. Samples were then transferred to 0.22 $\mu$m nylon filter centrifuge tubes (Corning Costar Spin-X Centrifuge Tubes) and quickly filtered by centrifugation. Samples were then stored at −20 °C until HPLC analysis, which was performed using the Rezex RoA (H+) organic acid column with 10 mM H$_2$SO$_4$ as the mobile phase.

### Computational and numerical methods
**Calculation of FBA solution and numerical derivatives.** The formulation of the FBA and FDM optimization problems is described in detail in Supplementary Note 1. The FBA calculations were performed in Python (version 3.9) using CVXPY (version 1.3.1)[66,67] and the GUROBI solver (GUROBIpy, version 9.5.2). In order to estimate the numerical derivatives reliably, we set the following GUROBI parameters: maximum iteration to 1000, "BarConvTol" and "BarQCPConvTol" to $10^{-12}$, "FeasibilityTol" and "OptimalityTol" to $10^{-9}$. Additionally, COBRApy version 0.26.3 was used to parse the metabolic models. Details on the implementation of FBA and of the functional decomposition are provided in Supplementary Notes 1 and 2, respectively. Additional analysis were performed in MATLAB, version R2015b.

**Hierarchical clustering.** The hierarchical clustering in Fig. 4a was performed on reactions whose fluxes were larger than $10^{-4}$ mmol/g$_{DW}$/h and only considering the metabolic functions associated with energy production, biosynthesis of amino acid and nucleotides, plus the mixed functional component; clusters were determined based on the cityblock distance among the functional shares $F_i^{(\gamma)}$. Full results including all the metabolic functions are reported in Supplementary Data 5.

**GO term-based decomposition.** We used the "biological process" terms to define the biological functions of *E. coli* proteins which were not categorized using the functional decomposition (Fig. 6b), as follows. We considered the average protein mass fractions between the reference condition and extreme C/R-limitations. The GO-term associated with the largest protein mass fractions was identified, and the genes associated with that GO-term were

assigned to the corresponding biological function. This process was iterated on the remainder of the genes until the largest protein mass fraction for each GO-term was less than 0.1%.

**Fit procedures.** Best fit parameters for the ATP maintenance flux were obtained by minimizing the squared residuals between the experimental and modeled glucose intake fluxes, $\sum_i (\Delta J_{\mathrm{glc},i})^2$, where $i$ indicates the samples. The best-fit parameters for the coarse-grained model, including the values of $v_C$ and $v_R$ in each growth condition, were obtained by minimizing the sum of two terms. The first term is the sum of squared residuals between the protein mass fractions of the five protein sectors $\alpha$ and the modeled mass fractions $\sum_{\alpha,i} (\Delta\phi_{\alpha,i})^2$; the second term are the squared residuals between modeled and experimental growth rates for each condition $\sum_i (\Delta\mu_i)^2$. The overall function to be optimized takes the form $\chi^2 = \sum_i (\Delta\phi_i)^2 + c\sum_i (\Delta\mu_i)^2$, where the scale factor $c$ is needed to compare the two terms since they have different physical units. We chose $c = 0.1\mathrm{h}^2$ so that the residuals of the growth rates are close to the typical experimental uncertainty on the growth rates (0.02/h or less).

### Reporting summary
Further information on research design is available in the Nature Portfolio Reporting Summary linked to this article.

## Data availability
Modified iML1515 *E. coli* metabolic models with biomass composition tailored to specific growth conditions are provided in Supplementary Data 1. Growth rate and flux data obtained in this work are provided in Supplementary Data 2. The functional decomposition of reaction fluxes and proteins in various conditions are provided in Supplementary Data 3 and 4. Energy, carbon and protein costs of biosynthetic activities are provided in Supplementary Data 6 and 8.

## Code availability
The Python code necessary to perform the functional decomposition on a given metabolic model is available on GitHub (https://github.com/ahoiching/FDM), and was deposited in ref. 68.

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

## Acknowledgements

MM thanks Andrea De Martino, Enzo Marinari, Cong Trinh and Martin Lercher for discussions. The work was supported by NSF grant no. MCB-1818384 and NIH grant no. R01GM095903 through TH.

## Author contributions

Conceptualization: M.M. and T.H. Methodology: M.M. Software: C.C. and M.M. Validation: C.C. and M.M. Formal analysis: M.M. Investigation: B.R.T. and H.O. (growth and flux measurement), C.C. and M.M. (data analysis). Data curation: C.C. and M.M. Writing - original draft: M.M. Writing - review and editing: M.M. and T.H. Visualization: C.C. and M.M. Supervision: M.M. and T.H. Project administration: M.M. Funding acquisition: T.H. All authors reviewed the results and approved the final version of the manuscript.

## Competing interests

The authors declare no competing interests.
