## [Peer Review File · Nature Communications]

Functional Decomposition of Metabolism allows a system-level quantification of fluxes and protein allocation towards specific metabolic functionsReviewer #1 (Remarks to the Author):

In this paper by Mori, Cheng and collaborators present a modeling framework (FDM) that aims to quantify the contribution of every metabolic reaction to metabolic functions. The modeling method is able to recapitulate known biological functions and modules. By using experimental measurements of growth, biomass composition, substrate consumption, fermentation by products and protein mass fractions they are able to quantify the contribution of each metabolic reaction and the allocation of the proteome to each cellular function. The modeling framework makes a contribution to the systems biology field by allowing to quantify the specific contributions of each process, however its application will be limited to have the quantitative proteomics data and the previously formulated constraints (such as the flux to the acetate pathway) probably limiting their application to other interesting biological questions, organisms or using different kinds of data sets. The mathematical formulation is quite difficult to understand, it would be desirable that the authors make an effort to present it in the main paper.

Major comments:

Additional constraints:

Jac difficult to interpret biologically? Is the basan proteome limitation not enough?

Figure 2B, how is it possible to operate a negative TCA cycle, this would mean carbon fixation and redox power consumption?

136-137 "...have to be evaluated on a case-by-case basis" This is a limitation of the extrapolation of this method to other systems. Could you be more specific on how a user of your method could deal with this in a completely different metabolic scenario? Or if it was to be implemented on a different organism?

Sign mismatched flux components should be better explained since it is quite confusing.

Figure 2E-F, in the fluxomics community the data is often presented as the percentage of the glucose intake. Here it may be useful to show how the "normalized" flux distribution is changing under different conditions. Since as it is now presented it is hard to grasp that the distribution of fluxes are changing, because most of them are increasing as the result of increased glucose intake.

Energy balance in the cell

209-210 unaccounted for energy (chemotaxis, metab futile cycle) how much is its contribution? KarrJ et al in whole-cell model of mycoplasma genitalium that accounts many more functions than just metabolism and biomass synthesis found a 44.3% of unaccounted for energy, however here it looks that it maybe (70-20/ 20) around 2.5X unaccounted for than accounted for energy. These numbers are quite difficult to understand, and contradictory to previous literature (even from Hwa's group). Why would the cell need to be more proteome efficient and do aerobic fermentation if it is under such energy abundance? Or are we so ignorant of the energy use of many other cellular processes? Or a third explanation is that something is wrong in the prediction of the production of such large amounts of ATP from respiration, aerobic fermentation and biosynthesis of precursors.

254 "suggesting that E. coli might be operating at reduced energetic efficiency in aerobic conditions" this claim needs further explanation/evidence

Further reviewing figure 4B, the ATP balance of biosynthesis precursors is not clear to me. For example, glutamine requires 1 ATP to be synthesized from glutamate and it has a negative value of ATP consumption (net ATP production if understood correctly). This could be a nice example to break down the calculations that yield a positive ATP outcome when intuitively or doing manual calculations one would expect a negative value.

Why only under anaerobic conditions the biosynthesis of precursors is actually consuming ATP and not producing it? As far as I understand, the stoichiometry of the biosynthetic reactions is the same under aerobic and anaerobic conditions. What could change is the fate of the NADH or NADPH produced on such reactions (if it is produced and not consumed) and if there is a surplus of

electrons under aerobic conditions it should be accounted as respiration and not as biosynthesis.
Or argue otherwise

Lane 315 - difference in growth rate has also been associated to the regulatory cost of expression unused or underused proteome (DOI: <https://doi.org/10.1371/journal.pcbi.1004998>)

Lane 318 - E. coli grows 20% slower, however its glucose intake is 2X than in aerobic conditions, this should also be analyzed in terms of biomass yield from carbon. Yes, the glucose uptake is not limiting when glucose is abundant, nor the glycolytic enzymes, nor the fermentative pathways. These has been extensively demonstrated elsewhere. But, energy is being produced at what rate compared to aerobic growth? The rates are key to understanding the differences in energy balance.

Would it be more precise to report the protein abundances in terms of mass or add information about the proteome size (e.g. mass/dry weight or mass/OD)? If proteome size were constant (growth-rate independent), there would be no conflict using mass fractions. However, proteome size is growth-rate dependent. The protein abundances expressed as fractions may occlude whether changes in it are due to the interest proteins increasing/decreasing their copy numbers or if their number remains constant and therefore it is the rest of the proteome that changes its composition. In two different growth conditions, if there were no growth-rate-dependent changes in copy numbers of (some) metabolic proteins, it would suggest that their efficiency is changing (increase/decrease).

Other metabolic reconstructions either constrained by the enzyme fraction of the proteome or that account for transcription/translation costs and macromolecular composition exist (e.g. ec models and ME models). Would using these models instead of iML1515 improve the decomposition of fluxes and/or could offer more precise flux predictions?

-Why was rich medium not included in the analysis?

Would it be more precise to report the protein abundances in terms of mass or add information about the proteome size (e.g. mass/dry weight or mass/OD)? If proteome size were constant (growth-rate independent), there would be no conflict using mass fractions. However, proteome size is growth-rate dependent. The protein abundances expressed as fractions may occlude whether changes in it are due to the interest proteins increasing/decreasing their copy numbers or if their number remains constant and therefore it is the rest of the proteome that changes its composition. In two different growth conditions, if there were no growth-rate-dependent changes in copy numbers of (some) metabolic proteins, it would suggest that their efficiency is changing (increase/decrease).

Minor comments:

"86-87 As described in Note 2, the set of the demand fluxes is not arbitrary, but is determined by the set of non-dimensionless constraints in the metabolic model" These non-dimensionless constraints should be explained since they are fundamental for understanding the modeling approach.

-In the iML1515 model, despite what the name suggests, there seems to be 1516 genes. Also, in the modified models in the supplementary files there are 1516 rows for genes. Why is it that in the main text the authors mention there are 1515 and not 1516 genes? (Line 155)

Redaction suggestions.

Main text:

Line 50: Change "functionality" to "function".

Line 73: Change "have" to "has".

Line 128: Change "allows us couple" to "allows us to couple".
Line 139: Change "differently" to "differentially".
Line 174: Change "Applying FDM" to "By applying FDM".
Line 214: Change "in reference condition" to "in the reference condition".
Line 278: Change "nucleotides biosynthesis" to "nucleotide biosynthesis".
Line 285: Change "the resulting energy cost were positive" to either "the resulting energy costs were positive" or "the resulting energy cost was positive".
Line 332: Change "in reference condition" to "in the reference condition".
Line 355: Change "in reference condition" to "in the reference condition".
Line 416: Change "hereogenously" to "heterogeneously".
Line 436: Change "these two term arise" to "these two terms arise".
Line 454: Change "functionality" to "function".

Sup. Text

Line 118: It says "iwth", it should say "with".
Line 227: Change "prescribes" to "prescribed".
Line 235: change "The resulting mass fractions were finally used to compute for each condition the demand fluxes for each biomass precursor" to "The resulting mass fractions were finally used to compute the demand fluxes for each biomass precursor in each of the conditions".
Line 292: Change "givens" to "given".
Line 366: Change "and to quantify how much does each reaction contribute to each function" to "and to quantify how much each reaction contributes to each function".
Line 509: The word "combining" is repeated.
Line 515: Is it "taking" or "perturbing"?

Reviewer #2 (Remarks to the Author):

Metabolic networks are highly interconnected as individual metabolic reactions can contribute to multiple metabolic functions, which complicates quantitative understanding of metabolism and its coordination. In this manuscript, Mori et al present a framework to decompose metabolic fluxes into different components, enabling quantification of the contribution of metabolic reactions to metabolic functions. Furthermore, by integrating proteomics data the framework allows for quantification of the amount of enzymes allocated to each metabolic function. This opens up a new direction in omics data analysis – in traditional analysis transcript/protein abundances of enzymes are usually summed up for pathways or GO-terms (ignoring the fact that an enzyme might be involved in multiple functions) but now the abundances can be summed up for metabolic functions. Overall, the study is timely and represents a significant advance in the field of systems biology, and the manuscript is well written. Please consider the relatively minor comments below.

To determine the ATP maintenance flux, the authors first minimized the glucose uptake and then minimized the difference between the measured and the simulated glucose fluxes. Is there any specific reason the authors did so? It would be more understandable to constrain the glucose uptake with the measured value and maximize the ATP maintenance flux, and this would lead to very similar results as the authors obtained.

Line 222-223: the conclusion is invalid and the sentence should be rephrased. It is impossible to distinguish the ATP generated by the biosynthesis of biomass building blocks from those by respiration and fermentation, and thus cells cannot exclusively utilize ATP generated by the biosynthesis of biomass building blocks for protein synthesis. One may only state like this: the ATP produced in conjunction with the biomass building blocks is close to the energetic demand associated with protein synthesis.

Line 224-225: this is also overstated. The ATP consumption reported here is only for protein synthesis, not for all known energy-consuming processes, and thus the ATP consumption for all processes should be higher. The ATP production by biosynthesis of biomass precursors is lower

than the ATP consumption by protein synthesis although the values are close, and it becomes much lower (empty blue bar in Figure 3) when glucose is less efficiently utilized. Therefore, ATP production by biosynthesis of biomass precursors cannot meet all ATP demands, i.e., the ATP fluxes associated to respiration and fermentation are needed. Please rephrase this sentence as well.

The authors estimated protein costs associated with energy and biomass production by integrating their framework with experimental proteomics data. This is very relevant to the work by integrating metabolic models with enzyme turnover numbers, which also estimated protein costs of synthesizing energy (PMID: 31405984) and recently amino acids (PMID: 35042799). It would be better to discuss or compare these two approaches. For example, one requires proteomics data while the other requires enzyme parameters. Moreover, it is worth emphasizing the consistent finding by the distinct approaches, i.e., high protein costs of synthesizing methionine, tryptophan and histidine in both *E. coli* and yeast.

Typos need to be fixed, and below are some but not all. Please carefully revise throughout the main text and the supplementary files.

Line 31: "study" -> "studying"

Line 54: "E. coli" -> "Escherichia coli"

Line 65: "model" -> "models"

Line 304: "Fig. 4E)" -> "(Fig. 4E)"

Line 309: "nor" -> "or"

Line 346: "Fig. 3B" -> "Fig. 3C"?

Line 442: "Fig. 5G" -> "Fig. 5E"?

Figure 3AB: the text "using NDH-II" is missing?

Figure 4 legend: "more expensive *that* most amino acids" -> "more expensive *than* most amino acids"

REVIEWER COMMENTS

Reviewer #1 (Remarks to the Author):

In this paper by Mori, Cheng and collaborators present a modeling framework (FDM) that aims to quantify the contribution of every metabolic reaction to metabolic functions. The modeling method is able to recapitulate known biological functions and modules. By using experimental measurements of growth, biomass composition, substrate consumption, fermentation by products and protein mass fractions they are able to quantify the contribution of each metabolic reaction and the allocation of the proteome to each cellular function. The modeling framework makes a contribution to the systems biology field by allowing to quantify the specific contributions of each process, however its application will be limited to have the quantitative proteomics data and the previously formulated constraints (such as the flux to the acetate pathway) probably limiting their application to other interesting biological questions, organisms or using different kinds of data sets. The mathematical formulation is quite difficult to understand, it would be desirable that the authors make an effort to present it in the main paper.

We sincerely appreciate the in-depth reading and critique of our work by the Reviewer. Prompted by their comments, we reworked the initial portion of the results section to provide a more comprehensive description of the FDM method, including the flux components and the sign-mismatch issue (see below). The revised section should provide a clearer overview of the method; however, due to the method's complexity, we still refer readers to the SI Note for a more detailed explanation.

Major comments:

Additional constraints:

Jac difficult to interpret biologically? Is the basan proteome limitation not enough?

Here (line 120 in the previous manuscript version), we did not intend to suggest that the widely studied phenomenon of acetate overflow lacks a biological interpretation. Rather, what requires a biological interpretation is the form of the flux component v_{ac} (or, equivalently, the flux mode ξ_{ac}) associated with acetate overflow. As explained below, the pathway described by these vectors, prior to any additional processing, cannot be easily interpreted due to the issues illustrated in Figure 2B (see below). We rephrased and expanded this sentence (lines 128-131) to avoid any ambiguity.

Figure 2B, how is it possible to operate a negative TCA cycle, this would mean carbon fixation and redox power consumption?

This is precisely what the form of the flux component v_{ac} would suggest. However, as the Reviewer suggests, this is not expected to happen. This is an example of the biological inconsistencies that can be generated by the flux decomposition in absence of any additional processing. Such inconsistencies are mathematically captured as "sign-mismatches" between the flux component and the overall flux direction. We hope that the revised text (lines 132-135) illustrates this point more clearly.

136-137 "...have to be evaluated on a case-by-case basis" This is a limitation of the extrapolation of this method to other systems. Could you be more specific on how a user of your method could deal with this in a completely different metabolic scenario? Or if it was to be implemented on a different organism?

We already provide a substantially different scenario in this manuscript when considering the case of anaerobically growing *E. coli*. Instead of a constraint on acetate excretion, our data shows excretion of succinate, which is not predicted by FBA in absence of further constraints. In order to faithfully capture the production of succinate, we constrained the corresponding excretion flux, which in turn led to the introduction of an additional flux component in the flux decomposition. This component presented problems similar to those obtained when constraining the acetate flux in the aerobic case - it ran fermentation in reverse to balance the cellular ATP, and hence produced succinate starting from ethanol and formate. We thus coupled this flux component with the one associated with fermentation (acetate/formate/ethanol-producing) to define a second fermentation pathway (acetate/formate/succinate-producing) with a lower energetic yield (see Supplementary Figure S5G).

We realize that, in the previous submission, this comment was not pertinent to this section of the manuscript. We thus moved this and a similar comment in the Result section to a dedicated paragraph in the Discussion section (lines 510-521) where we discuss the sign-mismatch problem, flux coupling and the application of the method to other organisms.

Sign mismatched flux components should be better explained since it is quite confusing.

We have elaborated more on the sign-mismatch problem in both the Results (lines 128-135) and in the Discussion (lines 510-521). We hope that our revised flow has made this important aspect of our method more accessible to readers.

Figure 2E-F, in the fluxomics community the data is often presented as the percentage of the glucose intake. Here it may be useful to show how the "normalized" flux distribution is changing under different conditions. Since as it is now presented it is hard to grasp that the distribution of fluxes are changing, because most of them are increasing as the result of increased glucose intake.

We agree that the overall growth-dependence of the fluxes makes it hard to appreciate the changes in the breakdown of the flux into the metabolic functions. We thus added a novel Supplementary Figure 2 in which we show the flux components from Fig. 2EF either as a fraction of the total reaction flux or as a fraction of the glucose intake flux.

Energy balance in the cell

209-210 unaccounted for energy (chemotaxis, metab futile cycle) how much is its contribution?

We thank the reviewer for the very pertinent question. The short answer is that it is either very small compared to protein synthesis (chemotaxis, but also mRNA synthesis), or unknown (futile cycles). In the previous version, this was only stated in a small paragraph with little to no explanation. We now include a detailed evaluation of the energy consumption

originating from flagella and mRNA turnover in Supplementary Note 4.5, which is referenced in a revised Main Text paragraph (lines 223-228).

Karr et al in whole-cell model of mycoplasma genitalium that accounts many more functions than just metabolism and biomass synthesis found a 44.3% of unaccounted for energy, however here it looks that it maybe (70-20/ 20) around 2.5X unaccounted for than accounted for energy. These numbers are quite difficult to understand, and contradictory to previous literature (even from Hwa's group).

From a qualitative point of view, we think that the results in Karr *et al.* agree with ours: energy is apparently produced well in excess compared to the estimated demands. However, a quantitative analysis of this discrepancy is not realistically feasible for several reasons. First and foremost, our manuscript and that by Karr *et al.* deal with extremely different organisms (*E. coli* and *M. genitalium*, respectively), for which we do not necessarily expect the same results in terms of energy balance and efficiency. Second, the whole-cell model employed in Karr *et al.* makes it very difficult to determine how the results depend on the assumptions for individual components of the model.

We address the consistency of our work with previous literature in response to the next comment below.

Why would the cell need to be more proteome efficient and do aerobic fermentation if it is under such energy abundance? Or are we so ignorant of the energy use of many other cellular processes?

These two questions on the origin of the observed excess of energy production are deeply interlinked and not necessarily mutually exclusive, so we will answer both at the same time.

We agree with the Reviewer that, if the observed excess in energy production weren't needed, then the cell could in principle reduce both the nutrient intake flux and the allocated proteome. Our analysis suggests two possibilities: (1) the cell is inefficiently producing energy via respiration, so that the actual energetic flux is lower than what can be estimated from the observed exchange fluxes, or that (2) the excess energy is consumed by unknown processes. We suspect that both (1) and (2) might hold true in the conditions studied here.

Regarding the proteome efficiency, the changes in NADH dehydrogenases' levels observed across conditions (Supplementary Figure S4B) suggest that ATP could be inefficiently produced at least in some growth conditions (i.e., when carbon source is not limiting). However, even assuming that the entire oxidative phosphorylation flux makes use of the least efficient enzyme NDH-II, the predicted reduction in ATP synthesis is not sufficient to completely balance the predicted energy synthesis and consumption. At the same time, while we couldn't pin down cellular processes able to explain the observed gap between produced and consumed energy, it is possible that this excess is consumed in futile cycles, but it is impossible to know for sure given the impossibility of directly evaluating their magnitude. Additionally, other unknown processes might require much more energy than anticipated.

The Reviewer suggested above a lack of consistency between these results and previous work from our group. In the picture developed in Basan et al. (PMID: 26632588), the proteome allocated to energy production (see Fig. 5C-D and Fig. 6D) must decrease at fast growth due to the increase in the protein share for biosynthetic activities (“biosynthesis” and “protein synthesis”) required to sustain the larger biosynthetic fluxes. The switch from respiration to fermentation is needed to accommodate this constraint.

We think that the presence of additional energy-consuming processes, or inefficiencies in energy generation, are broadly compatible with this picture. The presence of additional energy-consuming processes affected the switch between respiration and fermentation as predicted by the protein allocation constraints, suggesting that the constraints in proteome allocation might limit the energetic fluxes. Furthermore, we note that this transition is generically expected to happen if the ATP produced per unit of proteome by the fermentation pathway is larger compared to that of the respiration pathway. As can be seen in Fig. S4D, changing the efficiency of the electron transport chain does not alter significantly the ratio of the two efficiencies, because substrate-level phosphorylation only provides a small part of the ATP generated. Furthermore, the estimated energetic flux J_E is reduced by a similar amount. Therefore, changes in efficiency of the electron-transport chain do not directly affect the predicted transition between respiration and fermentation.

In general, our analysis highlights the need for a deeper, quantitative understanding of bacterial energetics and the regulation of energy producing processes. Prompted by the reviewer’s questions on this topic, we have changed the presentation of our results in various points of the Results section (lines 245-255, 262-265) and of the Discussion section (522-540) to better discuss the implications of our analysis.

Or a third explanation is that something is wrong in the prediction of the production of such large amounts of ATP from respiration, aerobic fermentation and biosynthesis of precursors.

Of course, we cannot completely exclude that we could be wrong in some of our predictions. Our confidence in the results stems from the wide variety of cross-checks performed in the manuscript. For instance, halving the ATP yield of respiration and fermentation (e.g., using the less efficient NADH dehydrogenase NDH-II, see Fig. 3) still leads to a total ATP production much higher than the known ATP costs. Furthermore, to make sure that our results on the energetics of the biosynthetic pathways were not affected by numerical errors, we performed extensive comparisons of the calculated carbon and energy costs of individual biosynthetic pathways with literature data. The results, summarized in Fig. 4 and Supp. Fig. S7, are in good agreement with previous analyses.

The flux decomposition is performed using the same code (which we provided on a GitHub repository) for all conditions. Given the cross-checks performed in Fig. 4 and Supp. Fig. S7, we are confident that our code performs well. On the other hand, approximations or mistakes in the FBA flux predictions to which FDM is applied (including both the values of the constrained fluxes or the biomass composition) might result in condition-specific oddities. During the revision process, we realized that the FBA solution for cells grown on acetate was not using the PTA/ACK node to uptake acetate, but rather the ACS (acetyl-CoA synthetase) reaction. Since the latter is more energetically expensive than the PTA/ACK pathway, both the energetic yield of respiration (shown in Fig. 4D) was underestimated and the energetic

costs for the biosynthetic reactions (in Fig. 4E and Supplementary File S8) were overestimated. This has been fixed in this version of the manuscript by allowing acetate intake via the PTA/ACK node, consistently with experimental evidence (PMID:28186174). Despite these changes, our results did not change qualitatively.

254 “suggesting that *E. coli* might be operating at reduced energetic efficiency in aerobic conditions” this claim needs further explanation/evidence

This sentence was not well connected to the previous results, and thus we agree that would have needed additional explanation. We expanded the relevant section (lines 262-265 and lines 277-279) to better relate the results on the energetics of anaerobic growth to those on aerobic growth and the efficiency of the NADH dehydrogenases.

Further reviewing figure 4B, the ATP balance of biosynthesis precursors is not clear to me. For example, glutamine requires 1 ATP to be synthesized from glutamate and it has a negative value of ATP consumption (net ATP production if understood correctly). This could be a nice example to break down the calculations that yield a positive ATP outcome when intuitively or doing manual calculations one would expect a negative value.

We thank the reviewer for giving us the opportunity to clarify some crucial details of our method. We now provide a complete breakdown of the glutamine case in Supplementary Figure S6 (referenced in the Main Text in lines 309-310). We will use this example to discuss the general properties of the method. Before going over the example, we summarize some key points here, which were not conveyed clearly in the previous version of the manuscript:

- Metabolic flux components represent complete mass-balanced pathways associated with a metabolic function, e.g., the biosynthesis of a given metabolite (such as glutamine) starting from the in-taken nutrients (e.g., glucose and ammonia). This is now made explicit in lines 101-104.
- Note that, because of mass-balance, electron carriers such as NAD/NADH, as well as ATP, are completely balanced across the network. (See also the comment below concerning the difference between aerobic and anaerobic energy costs.)
- Flux modes only differ from the corresponding flux components by an overall normalization factor; for biosynthetic pathways, flux modes are (arbitrarily) normalized relative to the flux of the synthesized metabolite. This simplifies the analysis of the pathways, even though only flux components are used to compute the functional decomposition of each reaction. This is also spelled out in lines 101-104.
- We define the energy cost associated with a flux mode as its net associated ATP hydrolysis flux. This flux matches the opposite of the flux through the ATP maintenance (ATPM) reaction necessary to mass-balance the flux mode. The yield is simply the opposite of the cost. We now define this terminology in lines 215-220.

Supplementary Figure S6 illustrates the flux mode for glutamine synthesis from glucose. In the case of glutamine, the net ATP produced from the central carbon pathways matches the consumption from glutamine synthetase (GLNS). However, the synthesis of one unit of glutamine from glucose also requires the reduction of three units of NAD to NADH. In order to fully balance the flux mode, NADH is oxidized by the electron transport chain, which in turn produces ATP. This production of ATP is balanced by ATPM. In this way, all metabolites

except for metabolites exchanged with the environment (e.g., glucose and ammonia) and glutamine (drained from the system by a sink reaction with unit flux, not shown in the diagram) are fully balanced. The flux through the ATPM reaction defines the overall energy yield of the flux mode, which in this case is 4.75.

The case for glutamate is identical, except that the reaction glutamine synthetase does not carry flux. Therefore, to preserve mass balance, the ATPM flux must increase by one unit, raising the energetic yield to 5.75. We hope that this example clarified the application of our method to the reviewer.

Why only under anaerobic conditions the biosynthesis of precursors is actually consuming ATP and not producing it? As far as I understand, the stoichiometry of the biosynthetic reactions is the same under aerobic and anaerobic conditions. What could change is the fate of the NADH or NADPH produced on such reactions (if it is produced and not consumed) and if there is a surplus of electrons under aerobic conditions it should be accounted as respiration and not as biosynthesis. Or argue otherwise

The reviewer's intuition is correct: in anaerobic conditions, electron-producing pathways no longer allow for the production of extra energy via the oxidative phosphorylation pathways and might instead cause a burden to the cell. We acknowledge that this was not explained in the previous version of the manuscript. Prompted by the Reviewer's comment, we performed a series of analyses to clarify *E. coli*'s energetics in anaerobic conditions.

As part of validating our approach, we had already shown (Supp. Fig. S7A) that the ATP cost of the biosynthetic reactions in aerobic conditions matched well those reported in classical physiology textbooks (Neidhardt et al., 1990). These costs include the ATP consumed because of the reduction of electron carriers. By separating the contribution of ATP and electron carriers, we were able to summarize the flow for the biomass biosynthesis in either aerobic (Supp. Fig. S5J) or anaerobic conditions (Supp. Fig. S5K; the corresponding numerical data in Supplementary File S6). As expected, the absence of oxygen forces a redistribution of the electron flow and a net ATP demand, rather than a net ATP production. This analysis is summarized in Main Text lines 280-288 and Supplementary Note 4.4.

Furthermore, as described in Supplementary Note 4.3, we were able to calculate the energetic equivalent of reduced electron carriers in anaerobic conditions by extending an analysis already performed for aerobic conditions (Supplementary Notes 4.2 and 4.3). As expected, the energetic equivalents are much smaller in anaerobic conditions (in absence of electron acceptors), and in some cases negative: oxidation of one NADH unit requires the hydrolysis of 0.25 ATP's, to be compared to the gain of 2 ATPs for the aerobic case.

Finally, the Reviewer states that the energy obtained from oxidative phosphorylation should be counted as "respiration". We want to stress that one of the main goals of our analysis was to determine the total costs associated with the synthesis of biomass building blocks. Whenever excess electrons are produced (or consumed) in conjunction with the building block, we associate the corresponding energy produced (or consumed) to the metabolite. In the example above, if (hypothetically) glutamine biosynthesis was to be reduced while

keeping everything else fixed, the overall energy synthesis flux would decrease because of the reduced amount of reduced electron carriers generated.

Accordingly, in our manuscript, the “respiration” energy share only refers to the energy share associated with the complete (mass-balanced) respiration of a glucose molecule to carbon dioxide, and not the total flux through the oxidative phosphorylation reactions (which is instead read off the FBA results). To avoid any confusion, we have clarified this in the Main Text, lines 185-186.

Lane 315 - difference in growth rate has also been associated to the regulatory cost of expression unused or underused proteome (DOI:

<https://doi.org/10.1371/journal.pcbi.1004998>)

While the underutilization of the proteome is unlikely to be the direct cause of the growth rate difference observed with varying glycolytic carbon sources, it can certainly affect it. We have included it and amended the text accordingly (lines 363-364).

Lane 318 - *E. coli* grows 20% slower, however its glucose intake is 2X than in aerobic conditions, this should also be analyzed in terms of biomass yield from carbon. Yes, the glucose uptake is not limiting when glucose is abundant, nor the glycolytic enzymes, nor the fermentative pathways. These has been extensively demonstrated elsewhere. But, energy is being produced at what rate compared to aerobic growth? The rates are key to understanding the differences in energy balance.

We thank the Reviewer for bringing up this important perspective, which we now discuss in Main Text, lines 352-361. The biomass yield in anaerobic growth is much lower compared to that in aerobic growth, i.e., the carbon intake is much higher; this is now shown in the newly added Supplementary Figure S5L. This larger intake flux is clearly driven by the much reduced efficiency for the energetic pathways. This can be seen by comparing Fig. 2E, top plot (the breakdown of the glucose intake flux in aerobic conditions) to Fig. S5M (same, but in anaerobic conditions). For a given growth rate, e.g., 0.7/h, the fraction of carbon intake associated to the biosynthesis of biomass building blocks (in blue) is similar between (carbon-limited) aerobic and anaerobic growth. On the other hand, the fraction associated with the energy-producing pathways is greatly different, with a much larger flux seen in the anaerobic case. Despite the lower ATP demand in anaerobic conditions, the drastically smaller energy production efficiency (ATP produced per glucose molecule) requires cells to uptake much larger amounts of glucose compared to cells grown at similar growth rates in aerobic conditions.

Would it be more precise to report the protein abundances in terms of mass or add information about the proteome size (e.g. mass/dry weight or mass/OD)? If proteome size were constant (growth-rate independent), there would be no conflict using mass fractions. However, proteome size is growth-rate dependent. The protein abundances expressed as fractions may occlude whether changes in it are due to the interest proteins increasing/decreasing their copy numbers or if their number remains constant and therefore it is the rest of the proteome that changes its composition. In two different growth conditions, if there were no growth-rate-dependent changes in copy numbers of (some) metabolic proteins, it would suggest that their efficiency is changing (increase/decrease).

The proteome takes an approximately constant fraction of the cellular dry mass, and the optical density is also proportional to the dry mass (PMID:26519362, PMID:36480614). Therefore, protein mass fractions are with good approximation proportional to protein mass/OD or protein mass/dry weight (but not to protein copy number per cell, since cellular volume varies greatly across conditions). Since protein mass fractions can be directly computed from mass spectrometry data without the need of additional measurements or information, we think it is more useful to report the results in terms of mass fractions rather than protein mass/OD or protein mass/gDW.

We have added a short remark in the text (lines 370-374) to stress that, since the proteome size is approximately constant across conditions, mass fractions are roughly proportional to protein concentrations.

Other metabolic reconstructions either constrained by the enzyme fraction of the proteome or that account for transcription/translation costs and macromolecular composition exist (e.g. ec models and ME models). Would using these models instead of iML1515 improve the decomposition of fluxes and/or could offer more precise flux predictions?

We thank the Reviewer for the very interesting question. The results obtained with our method are only as reliable as the modeled fluxes: if these do not match the experimental observations, then it does not make much sense to apply our method. Therefore, our approach was to consider the simplest modeling framework that produced flux predictions in quantitative agreement with the available data.

While models more complicated than "simple" metabolic networks, e.g., ME-models, can describe additional cellular processes and, possibly capture the interactions among metabolic fluxes and protein abundances, their use does not necessarily lead to a better agreement with experimental flux data. For example, the predictions for the acetate overflow with the ME-model shown in O'Brien et al., 2013 (PMID:24084808) are in quantitative (although not qualitative) disagreement with the experimental data. While the predictions can be improved by tweaking several model parameters (e.g. PMID:31158228), we are not sure that this would result in a better agreement between the modeled and experimental fluxes compared to the simpler approach followed here.

However, the application of the flux decomposition to more complex models, e.g. ME-models or RBA, would be potentially interesting, since it might allow to quantify the utilization of macromolecular machinery towards different metabolic functions, even though this would require carefully evaluating the constraints employed by these models. We now briefly discuss this exciting direction at the end of the Discussion section (lines 604-609).

-Why was rich medium not included in the analysis?

Because rich media conditions provide a host of additional challenges, both experimental and theoretical, compared to minimal media. First, it is challenging to reliably quantify the consumption rates of individual nutrients in complex media (amino acids, nucleotides, etc.). Even for the relatively simple case of glucose media supplemented with all 20 amino acids, measurement of intake rates has only been performed recently (PMID:32057959). Results of this study show that several amino acids, most notably serine, can be catabolized even in

the presence of other carbon sources (e.g., glucose). Properly modeling the metabolic fluxes with FBA would most likely require a plethora of additional constraints. In turn, this would complicate the application of FDM to the metabolic network. As this work aims at establishing the FDM method, we focused on the simpler case of minimal media, leaving the more challenging case of rich media for future work.

Would it be more precise to report the protein abundances in terms of mass or add information about the proteome size (e.g. mass/dry weight or mass/OD)? If proteome size were constant (growth-rate independent), there would be no conflict using mass fractions. However, proteome size is growth-rate dependent. The protein abundances expressed as fractions may occlude whether changes in it are due to the interest proteins increasing/decreasing their copy numbers or if their number remains constant and therefore it is the rest of the proteome that changes its composition. In two different growth conditions, if there were no growth-rate-dependent changes in copy numbers of (some) metabolic proteins, it would suggest that their efficiency is changing (increase/decrease).

This comment was addressed above.

Minor comments:

“86-87 As described in Note 2, the set of the demand fluxes is not arbitrary, but is determined by the set of non-dimensionless constraints in the metabolic model” These non-dimensionless constraints should be explained since they are fundamental for understanding the modeling approach.

We agree with the reviewer that this is a crucial point of our approach. In our case, these constraints are given by the demand fluxes of each biomass component, by the ATP maintenance flux, and by a single additional constraint - either the acetate excretion (aerobically) or the succinate excretion (anaerobically). We have clarified in lines 85-90 that the terms in the flux decomposition are set by the constraints at hand, and we state explicitly the set of constraints used in our application to *E. coli* later below (lines 182-184).

-In the iML1515 model, despite what the name suggests, there seems to be 1516 genes. Also, in the modified models in the supplementary files there are 1516 rows for genes. Why is it that in the main text the authors mention there are 1515 and not 1516 genes? (Line 155)

One of the 1516 genes (named s0001) appearing in the iML1515 model is actually not mapped to any real gene, and is instead assigned to all spontaneous reactions in the metabolic network, as seen in the associated BiGG entry:

<http://bigg.ucsd.edu/models/iML1515/genes/s0001>

Redaction suggestions.

Main text:

Line 50: Change “functionality” to “function”.

Line 73: Change “have” to “has”.

Line 128: Change “allows us couple” to “allows us to couple”.
Line 139: Change “differently” to “differentially”.
Line 174: Change “Applying FDM” to “By applying FDM”.
Line 214: Change “in reference condition” to “in the reference condition”.
Line 278: Change “nucleotides biosynthesis” to “nucleotide biosynthesis”.
Line 285: Change “the resulting energy cost were positive” to either “the resulting energy costs were positive” or “the resulting energy cost was positive”.
Line 332: Change “in reference condition” to “in the reference condition”.
Line 355: Change “in reference condition” to “in the reference condition”.
Line 416: Change “hereogenously” to “heterogeneously”.
Line 436: Change “these two term arise” to “these two terms arise”.
Line 454: Change “functionality” to “function”.

Sup. Text

Line 118: It says “iwth”, it should say “with”.
Line 227: Change “prescribes” to “prescribed”.
Line 235: change "The resulting mass fractions were finally used to compute for each condition the demand fluxes for each biomass precursor" to "The resulting mass fractions were finally used to compute the demand fluxes for each biomass precursor in each of the conditions".
Line 292: Change "givens" to "given".
Line 366: Change "and to quantify how much does each reaction contribute to each function" to "and to quantify how much each reaction contributes to each function".
Line 509: The word "combining" is repeated.
Line 515: Is it "taking" or "perturbing"?

We are grateful to the reviewer for providing us with these corrections. These and many other typos have been fixed in the reviewed version.

Reviewer #2 (Remarks to the Author):

Metabolic networks are highly interconnected as individual metabolic reactions can contribute to multiple metabolic functions, which complicates quantitative understanding of metabolism and its coordination. In this manuscript, Mori et al present a framework to decompose metabolic fluxes into different components, enabling quantification of the contribution of metabolic reactions to metabolic functions. Furthermore, by integrating proteomics data the framework allows for quantification of the amount of enzymes allocated to each metabolic function. This opens up a new direction in omics data analysis – in traditional analysis transcript/protein abundances of enzymes are usually summed up for pathways or GO-terms (ignoring the fact that an enzyme might be involved in multiple functions) but now the abundances can be summed up for metabolic functions. Overall, the study is timely and represents a significant advance in the field of systems biology, and the manuscript is well written. Please consider the relatively minor comments below.

We thank the Reviewer for the concise summary of the work and their positive opinion on our work.

To determine the ATP maintenance flux, the authors first minimized the glucose uptake and then minimized the difference between the measured and the simulated glucose fluxes. Is there any specific reason the authors did so? It would be more understandable to constrain the glucose uptake with the measured value and maximize the ATP maintenance flux, and this would lead to very similar results as the authors obtained.

From the point of view of FBA solutions, the two approaches are indeed similar (as long as only one carbon source is provided). We checked whether the results obtained with the two approaches were equivalent. As seen in the figure below, we obtain the exact same (to 6 digits) fit parameters for the case of C- and R-limitation, and very similar fit parameters for the case of anaerobic growth (-1.5 versus -1.9 mmol ATP/gDW for the slope, while the y-intercepts coincide). Such small differences do not affect our results significantly.

Figure 1: ATP maintenance flux obtained by maximizing the ATP flux with fixed glucose intake flux for aerobic (A) and anaerobic conditions (B). Symbols indicate the maximal ATP production flux for each glucose measurement, while the blue dashed lines indicate the best fitting (least square) linear models. For comparison, the best fit ATP maintenance relations with their confidence bands (from Fig. 2D and Supp. Fig. S5C).

The main reason underlying our approach is that, in the context of FDM, it is more intuitive to consider ATP as a constraint while optimizing for the glucose intake flux, and we did not want to change this procedure for the determination of the ATP parameters. However, we agree that the approach suggested by the Reviewer is more straightforward and mostly equivalent. We have noted it down in Supplementary Note 1, lines 293-296).

Line 222-223: the conclusion is invalid and the sentence should be rephrased. It is impossible to distinguish the ATP generated by the biosynthesis of biomass building blocks from those by respiration and fermentation, and thus cells cannot exclusively utilize ATP generated by the biosynthesis of biomass building blocks for protein synthesis. One may only state like this: the ATP produced in conjunction with the biomass building blocks is close to the energetic demand associated with protein synthesis.

We completely agree with the Reviewer, as this was the message we tried to convey. We revised the text to make it clearer (lines 240-242 in the revised version).

Line 224-225: this is also overstated. The ATP consumption reported here is only for protein synthesis, not for all known energy-consuming processes, and thus the ATP consumption for

all processes should be higher. The ATP production by biosynthesis of biomass precursors is lower than the ATP consumption by protein synthesis although the values are close, and it becomes much lower (empty blue bar in Figure 3) when glucose is less efficiently utilized. Therefore, ATP production by biosynthesis of biomass precursors cannot meet all ATP demands, i.e., the ATP fluxes associated to respiration and fermentation are needed. Please rephrase this sentence as well.

Also prompted by Reviewer #1, we expanded a section in the results (lines 223-228) and added a novel section in the Supplementary Notes (section 4.5) describing the known/unknown energetic costs. By far, protein synthesis is the dominant cost compared to other known and quantifiable processes such as mRNA turnover or chemotaxis.

We agree with the Reviewer that the previous version of the text suggested that the respiration and fermentation pathways are not needed at all for supporting the demand from protein synthesis, which is not what our results show. We have reformulated the text accordingly (lines 242-244), highlighting instead that only a small fraction of the flux through the respiration/fermentation pathways can be explained by the known costs.

The authors estimated protein costs associated with energy and biomass production by integrating their framework with experimental proteomics data. This is very relevant to the work by integrating metabolic models with enzyme turnover numbers, which also estimated protein costs of synthesizing energy (PMID: 31405984) and recently amino acids (PMID: 35042799). It would be better to discuss or compare these two approaches. For example, one requires proteomics data while the other requires enzyme parameters. Moreover, it is worth emphasizing the consistent finding by the distinct approaches, i.e., high protein costs of synthesizing methionine, tryptophan and histidine in both *E. coli* and yeast.

We appreciate the reviewer's feedback and have added a paragraph in the Discussion section (lines 594-609) to address the relationship between our work and other modeling efforts, including those mentioned by the Reviewer. We believe this addition is essential in providing necessary context to interested readers. Although we did not include details to avoid obstructing the flow of the paper, we now also present the results on yeast in line 418, in addition to those already included in the previous version of the manuscript (lines 409).

Typos need to be fixed, and below are some but not all. Please carefully revise throughout the main text and the supplementary files.

Line 31: "study" -> "studying"

Line 54: "E. coli" -> "Escherichia coli"

Line 65: "model" -> "models"

Line 304: "Fig. 4E)" -> "(Fig. 4E)"

Line 309: "nor" -> "or"

Line 346: "Fig. 3B" -> "Fig. 3C"?

Line 442: "Fig. 5G" -> "Fig. 5E"?

Figure 3AB: the text "using NDH-II" is missing?

Figure 4 legend: "more expensive *that* most amino acids" -> "more expensive *than* most amino acids"

We thank the reviewer for catching these mistakes. We have reviewed the manuscript and fixed typos to the best of our ability.

On our computers, the text “using NDH-II” in Figure 3AB is visible; see below for a screenshot of the pdf file.

Figure 2: Screenshot of Main Text Figure 3.

Reviewer #1 (Remarks to the Author):

I am satisfied with how my comments have been addressed, and I believe this new version of the article is much clearer and better.

However, there is one thing that still puzzles me - the fate of energy in aerobic conditions. As the authors stated, "However, there may be additional sources of ATP consumption, such as metabolic futile cycles, including those originating from membrane leakage, which are not fully understood and remain largely unknown [38]." Although this remains a challenge, it's possible that future investigations will solve this mystery.

Reviewing this paper has been both a challenge and a pleasure, I now have some hypotheses on where the energy flows that we may test in the lab.

Jose Utrilla - CCG UNAM (Mexico)
assisted by: David Hidalgo (pHD graduate)

Reviewer #2 (Remarks to the Author):

The authors have addressed all the comments, and I do not have any other comments.

Reviewer #3 (Remarks to the Author):

To the authors:

As a co-reviewer of the manuscript "Functional Decomposition of Metabolism allows a system-level quantification of fluxes and protein allocation towards specific metabolic functions" my comments and suggestions align to those of Dr. Utrilla. The authors have thoroughly addressed them and I believe the newest version of the manuscript has been improved, particularly in clarity.

-David Hidalgo - CCG UNAM